# Global and Local Topology-Aware Graph Generation via Dual Conditioning Diffusion

**Yuhang Xie** **Sinno Jialin Pan**
Department of Computer Science and Engineering
The Chinese University of Hong Kong, Hong Kong
`yuhangxie@link.cuhk.edu.hk, sinnopan@cuhk.edu.hk`

## Abstract

Graph generation plays an important role in various domains such as molecular design, protein prediction, and drug discovery. However, generating graph-structured data poses challenges due to the complex dependencies inherent in graphs, spanning from intricate local substructures to broad global topologies. Although recent advances in graph-generative models have made notable progress, traditional node-level generative paradigms may have difficulty simultaneously capturing the multiscale dependencies in graphs. To address these challenges, we propose a unified latent diffusion model that jointly learns local and global topological information, enabling effective and efficient graph generation. Besides, our approach introduces a dual conditioning mechanism designed to promote dynamic interaction between local and global information, equipping the generative model with global and local awareness to better capture the coupled dependencies within graphs. Our method can largely promote the joint modeling of global and local information and substantially improve the quality of the generated graphs. Extensive experiments consistently demonstrate the effectiveness of our method.

## 1 Introduction

Graph-structural data provide powerful representations for describing objects and their relationships in diverse real-world domains, such as social networks (Singh et al., 2024), biological systems (Liu et al., 2024; Zang & Wang, 2020), and traffic networks (Cui et al., 2019). Among the core tasks in graph machine learning, graph generation has emerged as a powerful tool with significant potential in diverse applications such as molecular structure modeling (Xu et al., 2023; Vignac et al., 2022; Luo et al., 2023), circuit design (Shahane et al., 2023), and source code generation (Dai et al., 2018).

Currently, advances in deep models, including generative adversarial networks (Goodfellow et al., 2020), diffusion-based models (Ho et al., 2020; Jo et al., 2022; Luo et al., 2023), and flow-based methods (Chen et al., 2018; Bengio et al., 2023; Zhang et al., 2024b), have paved the way for effective graph generation. Unlike densely distributed image data, graph data can be extremely sparse and often exhibits complex topological information, especially the coupled local and global dependencies within distinct structures (Guo & Zhao, 2022). For instance, social networks exhibit intricate dependencies both within individual communities and between different communities (Singh et al., 2024); likewise, molecular graphs demonstrate complex relationships both within and among substructures (Zhao et al., 2012). Specifically, the local structure of a molecule, such as the specific functional groups, plays a crucial role in determining its chemical reactivity. However, the global information, including the overall structural topology and the spatial distribution of functional groups, also significantly impacts its overall properties and functional performance (Livingstone, 2000). Therefore, the intrinsic and distinct dependencies present in graphs highlight the importance of integrating both global and local information to enable a more precise graph generation process.

Recently, some effective models have incorporated global information to enhance the graph generation process. For instance, SubgDiff (Zhang et al., 2024a) introduces a subgraph prediction module that integrates substructure information into diffusion models, thereby improving molecular representation learning. Similarly, Graphusion (Yang et al., 2024) leverages graph-level pseudo-labels derived from clustering algorithms to provide informative guidance during the generation process.

However, these approaches may overlook the dependencies among substructures and have limitations in modeling the joint distribution of global and local information. This raises a natural and important question: *How can we design a unified generative model that can dynamically capture both global and local topological information to enable effective graph generation?*

To answer this question, we propose a unified dual conditioning latent diffusion model, termed **DualDiff**, which is capable of effectively modeling the joint distribution of global and local information, facilitating a robust and stable generation process. Concretely, DualDiff first maps the original graph space and diverse graph features into a unified latent space via a pretrained graph autoencoder, then it extracts the global information according to the local representations. Subsequently, DualDiff employs a two-branch diffusion process to learn topological dependencies at both the node and subgraph levels within a unified framework. To further advance the joint modeling of global and local information, a dual conditioning mechanism is introduced to promote interaction between these two branches, wherein global and local information are alternately utilized as conditions to guide the learning process of the complementary branch.

By leveraging the two-branch diffusion process and dual conditioning, DualDiff can be equipped with both global and local topological awareness, enabling it to dynamically capture the dependencies within the coarse-grained global structures and the fine-grained local details. Experimentally, we conduct comprehensive evaluations on graph generation benchmarks, further demonstrating the effectiveness of our proposed method. Our main contributions are summarized as follows:

- We highlight the limitations of conventional graph generative models in capturing the joint distribution of global and local topological information within graphs. To this end, we propose DualDiff, an effective latent diffusion-based generative model capable of simultaneously learning the dependencies inherent in both global and local structures.
- By incorporating the proposed two-branch diffusion process and dual conditioning mechanism, DualDiff can effectively capture the joint global and local topological dependencies.
- Extensive experiments across diverse benchmarks, including generic graphs and real-world molecular datasets, demonstrate that DualDiff consistently outperforms existing methods, largely illustrating the effectiveness of our proposed method.

## 2 RELATED WORK

**Generative Models on Graph.** Graph generative models have been widely adopted across various domains. Early approaches include graph-based variants of variational autoencoders (VAEs) (Kingma et al., 2013; Kipf & Welling, 2016; Jin et al., 2018) and generative adversarial networks (GANs) (Goodfellow et al., 2020; De Cao & Kipf, 2018; Krawczuk et al., 2020), flow-based models (Lippe & Gavves, 2021; Luo et al., 2021), and autoregressive models (You et al., 2018; Liao et al., 2019). More recently, diffusion-based models have demonstrated state-of-the-art performance on graph generation tasks (Jo et al., 2022; Qiang et al., 2023; Yan et al., 2023; Luo et al., 2023). Despite these advancements, effectively modeling the complex and coupled topological dependencies inherent in graphs remains a significant challenge. To address this limitation, we propose a unified framework, DualDiff, designed to simultaneously capture both global and local topological information, thereby facilitating a more efficient and expressive graph generation process.

**Latent Graph Diffusion Model.** Latent diffusion model (LDM) (Rombach et al., 2022; Podell et al., 2023) leverages the idea of performing the diffusion process in a lower-dimensional and compact latent space instead of the original data space, which greatly improves efficiency and scalability. Building on the success of LDM in images, GEOLDM (Xu et al., 2023), EDM-SYCO (Ketata et al., 2025), and LGD (Zhou et al., 2024) extend LDM to molecular and graph generation tasks. They embed the graph structures and features into a latent space by a pretrained autoencoder and conduct a diffusion process in the latent space, exhibiting competitive performance and high computational efficiency. However, these methods still rely on node-level generative methods, which treat each node as an independent entity and may overlook dependency within substructures.

**Conditioning on Diffusion Model.** Conditioning (Ho et al., 2022; Nichol & Dhariwal, 2021) is a widely used technique for guiding diffusion models to generate samples in a targeted and meaningful manner. Traditional conditioning methods typically incorporate external information, such as

class labels or auxiliary inputs like low-resolution images (Zhang et al., 2023). More recently, self-conditioning (Chen et al., 2023) allows models to leverage their own previous predictions, resulting in notable improvements in sample quality. In addition, GPrinFlowNet (Mo et al., 2024) introduces a low-to-high frequency generation curriculum, effectively preserving semantic information and achieving high-quality conditional graph generation. Distinct from these prior approaches, we propose a dual conditioning mechanism that facilitates interaction between global and local information. This design endows our model with both global and local topological awareness, enabling more effective learning of the diverse dependencies present in graph-structured data.

## 3 PRELIMINARY

**Diffusion Models.** Diffusion models have demonstrated strong capabilities in modeling complex data distributions (Sohl-Dickstein et al., 2015; Ho et al., 2020). In these models, the forward process gradually corrupts a data sample $\mathbf{x}_0$ by adding Gaussian noise over $T$ steps, producing a sequence $\mathbf{x}_t$ according to $q(\mathbf{x}_t|\mathbf{x}_{t-1}) = \mathcal{N}(\mathbf{x}_t; \sqrt{1-\beta_t}\mathbf{x}_{t-1}, \beta_t\boldsymbol{I})$, where $\beta_t$ is the noise schedule. The objective of diffusion is to learn the reverse process $p_\theta(\mathbf{x}_{t-1}|\mathbf{x}_t)$, which denoises $\mathbf{x}_t$ to recover the original distribution. Elucidated Diffusion Models (EDMs) (Karras et al., 2022) extend this framework by generalizing the noise schedule and reverse process parameterization. In EDM, the model is trained to predict clean data from noisy inputs across different noise levels. Specifically, given a noise level $\sigma$, the denoising loss is defined as $\mathbb{E}_{p_{data}(\mathbf{x})p_\sigma(\tilde{\mathbf{x}}|\mathbf{x})}||D_\theta(\tilde{\mathbf{x}}, \sigma) - \mathbf{x}||$, where $D_\theta(\tilde{\mathbf{x}}, \sigma)$ is the denoising network, and $\tilde{\mathbf{x}}$ is the noisy data sampled as $p_\sigma(\tilde{\mathbf{x}}|\mathbf{x}) = \mathcal{N}(\tilde{\mathbf{x}}; \mathbf{x}, \sigma^2\boldsymbol{I})$. During reverse process, the learned denoising network approximates the score function $\nabla\mathbf{x}\log p(\mathbf{x}; \sigma) = (D_\theta(\tilde{\mathbf{x}}, \sigma) - \mathbf{x})/\sigma^2$, enabling sample generation via numerical solvers as described in (Song et al., 2021).

**Self-Conditioning.** Self-conditioning is a tool to improve diffusion models by allowing the denoising network to utilize its own previous predictions as conditions (Chen et al., 2023). Specifically, self-conditioning feeds the previously predicted clean sample $\hat{\mathbf{x}}_0$ back into the model as an additional input, which can largely enhance sample quality and consistency with minimal computational overhead. During training, self-conditioning is applied with a probability $p$ to prevent the model from over-relying on self-conditioned inputs, thereby improving the robustness of the learning process.

## 4 METHODOLOGY

**Problem Definition.** In this paper, we consider graph generation from scratch. Let $N$ denote the number of nodes in a given graph, then a graph can be defined as $\mathcal{G} = (\boldsymbol{H}, \boldsymbol{A})$, where $\boldsymbol{H} \in \mathbb{R}^{N \times d_h}$ denotes the node feature matrix, and $\boldsymbol{A} \in \mathbb{R}^{N \times N}$ represents the adjacency matrix. For tasks related to biology and chemistry, the 3D coordinate matrix $\boldsymbol{X} \in \mathbb{R}^{N \times 3}$ can also be available. The graph autoencoder utilized in our latent diffusion framework consists of an encoder $\mathcal{E}_\phi$ and a decoder $\mathcal{D}_\psi$. The encoder $\mathcal{E}_\phi$ transforms $\mathcal{G}$ into a latent representation $\boldsymbol{Z} \in \mathbb{R}^{N \times d}$. We use $\boldsymbol{Z}_l \in \mathbb{R}^{N \times d}$ to denote the local information and $\boldsymbol{Z}_g \in \mathbb{R}^{K \times d}$ to represent the global information, where $\boldsymbol{Z}_g$ is obtained by applying clustering to $\boldsymbol{Z}_l$ and $K$ is the number of clusters. By employing advanced graph clustering methods, $\boldsymbol{Z}_g$ can effectively encode important global information within graphs. Our objective is to employ a unified latent diffusion model to model the joint distribution $p(\boldsymbol{Z}_l, \boldsymbol{Z}_g)$.

**Overview.** The primary motivation behind DualDiff is to endow the model with both global and local structural awareness. For example, in molecular generation tasks, DualDiff can accurately capture local details, such as specific functional groups or molecular fragments, while simultaneously considering the molecule's overall topology and the distribution of substructures. The architecture of DualDiff is depicted in Figure 1. We first describe the construction of the autoencoder and global information extraction in Section 4.1. Then, the details of DualDiff will be illustrated in Section 4.2. Finally, we summarize the training and sampling scheme of DualDiff in Section 4.3.

### 4.1 AUTOENCODER CONSTRUCTION AND GLOBAL INFORMATION EXTRACTION

**AutoEncoder.** In graph-structured data, there often exist many meaningful features, such as molecular conformation positions and charges. To fully utilize these informative features, we leverage the Latent Graph Generation (LGD) paradigms to map graphs and the features into a unified

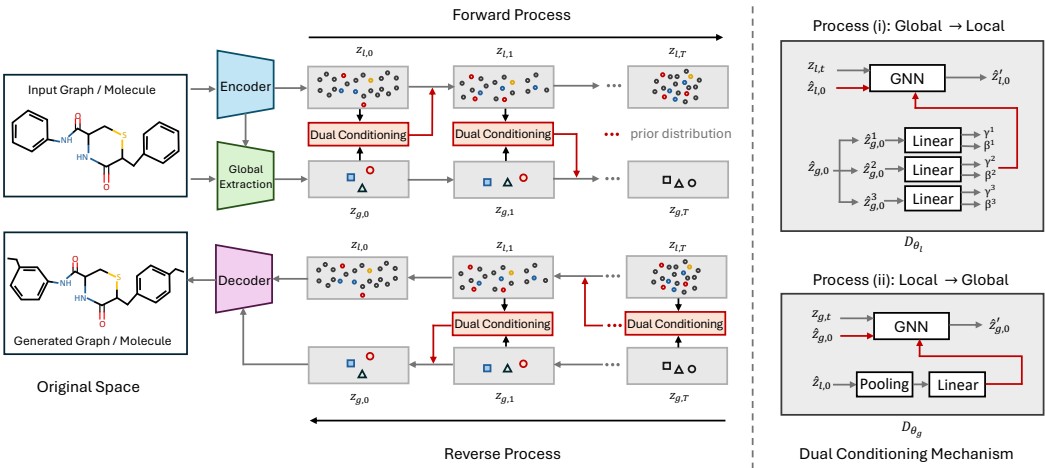

Figure 1: The workflow of DualDiff (**Left**) and details of the dual conditioning mechanism (**Right**).

latent space, thereby integrating rich feature information to facilitate generation. Specifically, we follow the settings in previous work (Rombach et al., 2022; Xu et al., 2023; Ketata et al., 2025) and pretrain an autoencoder to enable the transformation between graph space and latent space. The encoding and decoding processes can be formulated by $q_\phi(\boldsymbol{Z}|\mathcal{G}) = \mathcal{N}(\mathcal{E}_\phi(\mathcal{G}), \sigma_0 \boldsymbol{I})$ and $p_\psi(\mathcal{G}|\boldsymbol{Z}_l, \boldsymbol{Z}_g)$.

Regarding the encoder architecture, in chemical and biological domains, nodes are generally endowed with 3D coordinate information, which inherently encodes spatial adjacency relationships. To exploit this property, we incorporate equivariance into $\mathcal{E}_\phi$ by parameterizing the encoder with Equivariant Graph Neural Networks (EGNNs) (Satorras et al., 2021), thereby ensuring that the induced latent representations are SE(3)-equivariant. For general graph-structured data lacking explicit coordinate information, $\mathcal{E}_\phi$ can be instantiated with various established graph neural network architectures, such as GIN (Xu et al., 2019), GCN (Kipf & Welling, 2017), and GAT (Veličković et al., 2018), to effectively facilitate message passing among nodes. To reconstruct the original graph, our decoder exploits both global and local structural information, which can be implemented as lightweight MLPs. The overall framework is summarized as follows:

$$\boldsymbol{Z} = \mathcal{E}_\phi(\mathcal{G}) \implies \begin{cases} \boldsymbol{Z}_l = \boldsymbol{Z} + \sigma_0 \boldsymbol{I} \\ \boldsymbol{Z}_g = \text{GlobalExtraction}(\boldsymbol{Z}, \mathcal{G}) \end{cases} \implies \hat{\mathcal{G}} = \mathcal{D}_\psi(\boldsymbol{Z}_l, \boldsymbol{Z}_g), \tag{1}$$

where $\text{GlobalExtraction}$ represents the operation of global information extraction, which will be explained later. Then, the whole framework can be effectively optimized by:

$$\mathcal{L}_{rec} = -\mathbb{E}_{q(\mathcal{G})q_\phi(\boldsymbol{Z}|\mathcal{G})} \left[ p_\psi(\mathcal{G}|\boldsymbol{Z}_l, \boldsymbol{Z}_g) \right]. \tag{2}$$

In practice, the reconstruction loss can be calculated as the $L_2$ norm or the cross-entropy function. More details of the autoencoder and latent space can be found in Appendix A.1.

**Global Information Extraction.** Recent studies have demonstrated the critical role of global information in graph representation learning, particularly in applications like molecular property prediction and social network analysis (Yang et al., 2019; Fang et al., 2022). By incorporating global information, models can capture long-range dependencies and holistic structural patterns that cannot be fully represented by local features, thus largely improving the representation capability.

To extract global topological information from a graph, we leverage graph clustering methods (Schaeffer, 2007), which are powerful tools for uncovering intrinsic structural patterns within graphs. To obtain topologically-enhanced global information, we first employ different clustering strategies based on the nature of the graph. Specifically, for molecular graphs, we apply the $K$-means algorithm in the atom coordinate space to generate geometry-enhanced class labels. For generic graphs, we utilize spectral clustering (Von Luxburg, 2007) that partitions nodes according to the eigenvectors of the graph Laplacian matrix. Following the clustering process, we aggregate node features within each cluster using a pooling operation to derive cluster-level embeddings. The overall process of

global information extraction can be formulated as follows:

$$\boldsymbol{S}_g = \text{Clustering}(\mathcal{G}) \implies \boldsymbol{Z}_g = \text{Pooling}(\boldsymbol{S}_g, \boldsymbol{Z}) \in \mathbb{R}^{K \times d}, \tag{3}$$

where $\boldsymbol{S}_g \in \{0,1\}^{N \times K}$ is the affiliation matrix indicating the cluster assignment of each node. By introducing topologically-enhanced global information, our model can effectively capture dependencies among global structures and guide the generation of local details more efficiently. We provide more detailed explanations regarding the effectiveness of the clustering methods in Appendix C.8.

## 4.2 Two-Branch Diffusion with Dual Conditioning

To integrate global and local information, we propose a unified diffusion framework that explicitly models both node-level and cluster-level dependencies. Furthermore, a dual conditioning mechanism is proposed to facilitate effective information exchange between global and local information, thereby promoting the model to capture the joint distribution of global and local representations.

**Two-Branch Diffusion on Global and Local Information.** Given that global structures and local details within a graph generally exhibit distinct topological characteristics, we introduce a unified two-branch diffusion process operating at both the node and cluster levels, which enables the model to effectively capture local and global information, respectively. Following (Song et al., 2021), the forward process of this diffusion can be formulated as a system of stochastic differential equations (SDEs) defined in the latent space, as follows:

$$\begin{cases} d\boldsymbol{Z}_{l,t} = f_{l,t}(\boldsymbol{Z}_{l,t})dt + s_{l,t}d\boldsymbol{W}_{l,t}, \\ d\boldsymbol{Z}_{g,t} = f_{g,t}(\boldsymbol{Z}_{g,t})dt + s_{g,t}d\boldsymbol{W}_{g,t}, \end{cases} \tag{4}$$

where $f_{l,t}(\cdot)$ and $f_{g,t}(\cdot)$ denote the drift coefficients, $s_{l,t}$ and $s_{g,t}$ are the diffusion coefficients, which are typically formulated as deterministic functions of time $t$, $\boldsymbol{W}_{l,t}$ and $\boldsymbol{W}_{g,t}$ represent the standard Wiener processes. Correspondingly, the reverse-time SDE system of (4) can be given by:

$$\begin{cases} d\bar{\boldsymbol{Z}}_{l,t} = \left( f_{l,t}(\bar{\boldsymbol{Z}}_{l,t}) - s_{l,t}^2 \nabla_{\boldsymbol{Z}_l} \log p_t(\bar{\boldsymbol{Z}}_{l,t}) \right) d\bar{t} + s_{l,t}d\bar{\boldsymbol{W}}_{l,t}, \\ d\bar{\boldsymbol{Z}}_{g,t} = \left( f_{g,t}(\bar{\boldsymbol{Z}}_{g,t}) - s_{g,t}^2 \nabla_{\boldsymbol{Z}_g} \log p_t(\bar{\boldsymbol{Z}}_{g,t}) \right) d\bar{t} + s_{g,t}d\bar{\boldsymbol{W}}_{g,t}, \end{cases} \tag{5}$$

where $d\bar{t} = -dt$ is the negative infinitesimal time step, $\bar{\boldsymbol{W}}_{l,t}$ and $\bar{\boldsymbol{W}}_{g,t}$ represent the reverse-time Wiener processes, and $\nabla \log p_t(\cdot)$ is the score function. Under the EDM framework, we can set $f_{l,t}(\boldsymbol{Z}_{l,t}) = \boldsymbol{0}$, $f_{g,t}(\boldsymbol{Z}_{g,t}) = \boldsymbol{0}$, and $s_{l,t} = s_{g,t} = \sqrt{2t}$. To predict the clean global and local latent embeddings, we employ two separate denoising networks, $D_{\theta_l}$ and $D_{\theta_g}$, which can be instantiated as GNNs in practice. Then, the overall training objective can be formulated as:

$$\mathbb{E}_{(\boldsymbol{Z}_{l,0}, \boldsymbol{Z}_{g,0}) \sim q_\phi(\cdot|\mathcal{G}) p_\sigma(\tilde{\boldsymbol{Z}}_l, \tilde{\boldsymbol{Z}}_g | \boldsymbol{Z}_{l,0}, \boldsymbol{Z}_{g,0})} [\|D_{\theta_l}(\tilde{\boldsymbol{Z}}_l, \sigma) - \boldsymbol{Z}_{l,0}\|^2 + \|D_{\theta_g}(\tilde{\boldsymbol{Z}}_g, \sigma) - \boldsymbol{Z}_{g,0}\|^2]. \tag{6}$$

In the proposed two-branch diffusion process, the denoising networks are designed to capture node-level and cluster-level dependencies independently. However, this architecture has limitations in modeling the joint distribution between global and local information. To further promote dynamic information exchange and enable joint modeling of global and local representations, we introduce a dual conditioning mechanism, as described in the following part.

**Dual Conditioning Mechanism.** In this part, we propose a dual conditioning mechanism to achieve dynamic interactions between different topological knowledge, thereby equipping the model with comprehensive global and local topological awareness. According to the conditional probability formulation, the joint distribution of global and local information can be expressed as:

$$p(\boldsymbol{Z}_l, \boldsymbol{Z}_g) = p(\boldsymbol{Z}_l|\boldsymbol{Z}_g)p(\boldsymbol{Z}_g) = p(\boldsymbol{Z}_g|\boldsymbol{Z}_l)p(\boldsymbol{Z}_l). \tag{7}$$

This equality implies that modeling the joint distribution of global and local information can be decomposed as two complementary processes: (i) introducing global information into the modeling of local details, associated with $p(\boldsymbol{Z}_l|\boldsymbol{Z}_g)$; and (ii) incorporating local information into the modeling of global structures, associated with $p(\boldsymbol{Z}_g|\boldsymbol{Z}_l)$.

Specifically, we first follow the self-conditioning and obtain the previously estimated $\hat{Z}_{l,0}$ and $\hat{Z}_{g,0}$ produced by $D_{\theta_l}$ and $D_{\theta_g}$, respectively. Then, we combine local and global information as conditions to alternately guide the local and global diffusion process. Let $C_l$ and $C_g$ respectively denote the conditions introduced to the local and global diffusion processes, then they can be given by:

$$(C_l, C_g) = \begin{cases} ((\hat{Z}_{l,0}, \hat{Z}_{g,0}), \mathbf{0}), & \text{with prob. } p, \quad \color{gray}{\text{related to } p(Z_l|Z_g)p(Z_g)} \\ (\mathbf{0}, (\hat{Z}_{l,0}, \hat{Z}_{g,0})), & \text{with prob. } 1-p, \quad \color{gray}{\text{related to } p(Z_l)p(Z_g|Z_l)} \end{cases} \tag{8}$$

where $p$ denotes the frequency of process (i). In detail, during process (i), to match the first decomposition $p(Z_l|Z_g)p(Z_g)$ in (7), we use $C_l = (\hat{Z}_{l,0}, \hat{Z}_{g,0})$ to incorporate both self-condition and global information to guide the learning of $Z_l$, thereby modeling $p(Z_l|Z_g)$. Meanwhile, $C_g$ is set to $\mathbf{0}$ to align with $p(Z_g)$. Similarly, we can obtain $(C_l, C_g) = (\mathbf{0}, (\hat{Z}_{l,0}, \hat{Z}_{g,0}))$ during process (ii). More interestingly, this strategy naturally matches the self-conditioning, where the condition is set to zero with a certain probability to enhance the robustness of the generation process, further guaranteeing the effectiveness of the proposed dual conditioning. A comparison of different conditioning methods is provided in Figure 2. Next, we will present a detailed description of the two processes used in the dual conditioning mechanism, as illustrated in the right part of Figure 1.

*(i) Incorporating global information into the local structure.* According to the conditional probability $p(Z_l|Z_g)$, we can introduce the global information to promote the generation process of local topologies. Concretely, inspired by FiLM (Perez et al., 2018), we first pass the predicted global representation $\hat{Z}_{g,0}$ through linear transformation to yield scale and shift parameters, $\gamma^i \in \mathbb{R}^d$ and $\beta^i \in \mathbb{R}^d$ $(i = 1, \ldots, K)$. Next, each node is assigned to a cluster $y_i$ based on the similarity between $\hat{Z}_{l,0}$ and $\hat{Z}_{g,0}$.

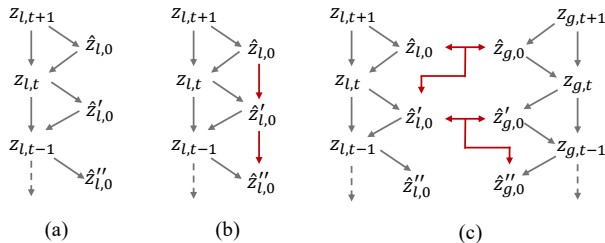

Figure 2: Comparison between different conditioning methods during the reverse process. (a) diffusion without conditioning; (b) self-conditioning; (c) dual conditioning.

Finally, the scale and shift parameters associated with each cluster are utilized to steer the distribution of the final output $\hat{Z}'_{l,0}$, thereby integrating global awareness into the modeling of local structures. Formally, we have:

$$\hat{Z}'^{,i}_{l,0} = \gamma^{y_i} \odot \hat{Z}^i_{l,0} + \beta^{y_i}, \text{ where } y_i = \arg\max_{j=1,\ldots,K} \text{sim}(\hat{Z}^i_{l,0}, \hat{Z}^j_{g,0}), \ \hat{Z}_{l,0} = \text{GNN}(Z_{l,t}, \hat{Z}_{l,0}, \sigma_t). \tag{9}$$

*(ii) Incorporating local information into the global topology.* In addition to process (i), local information can also be leveraged to help predict global structures, thus modeling $p(Z_g|Z_l)$. Specifically, the estimated local details $\hat{Z}_{l,0}$ can be processed through message passing (MP) and pooling (Pool) operations to obtain a low-dimensional global condition $C$. This condition is then concatenated with the self-condition $\hat{Z}_{g,0}$ as the final condition of the denoising network. Formally, we have:

$$C = \text{Linear}(\text{Pool}(\text{MP}(\hat{Z}_{l,0}))) \Rightarrow C = \text{Concat}(\hat{Z}_{g,0}, C) \Rightarrow \hat{Z}'_{g,0} = \text{GNN}(Z_{g,t}, C, \sigma_t). \tag{10}$$

In conclusion, the dual conditioning mechanism can naturally integrate processes (i) and (ii), enabling dynamic interaction between global and local information and equipping DualDiff with global and local topological awareness, which largely promotes the effectiveness of the generation process. We also include more details of the dual conditioning mechanism in Appendix A.4.

## 4.3 TRAINING AND SAMPLING

In this section, we present the detailed training and sampling procedures for DualDiff. Consistent with the latent diffusion model (LDM) framework, we employ a two-stage training strategy: an autoencoder is first pretrained, and remains fixed throughout both the diffusion training and sampling phases. To facilitate a more stable sampling process, we draw inspiration from the server-client communication paradigm, where the server updates global parameters after accumulating several updates from clients to promote training stability (McMahan et al., 2017). We analogize the global

clusters to the role of the global server and their corresponding nodes to the local clients. Correspondingly, process (i) and process (ii) can be viewed as local update and global aggregation, respectively. Therefore, we alternate $m$ steps of process (i) with a single step of process (ii) to enhance the stability of the sampling process. Besides, this alternating strategy can significantly improve sampling stability and model performance, compared to executing two processes simultaneously. Further details regarding the effectiveness of dual conditioning and the corresponding theoretical insights can be found in Appendix D. Finally, with the integration of dual conditioning, the overall loss objectives are updated as follows:

$$\mathbb{E}_{(\boldsymbol{Z}_{l,0}, \boldsymbol{Z}_{g,0}) \sim q_\phi(\cdot|\mathcal{G})p_\sigma(\tilde{\boldsymbol{Z}}_l, \tilde{\boldsymbol{Z}}_g | \boldsymbol{Z}_{l,0}, \boldsymbol{Z}_{g,0})} [\|D_{\theta_l}(\tilde{\boldsymbol{Z}}_l, \boldsymbol{C}_l, \sigma) - \boldsymbol{Z}_{l,0}\|^2 + \|D_{\theta_g}(\tilde{\boldsymbol{Z}}_g, \boldsymbol{C}_g, \sigma) - \boldsymbol{Z}_{g,0}\|^2]. \quad (11)$$

Overall, the training and sampling process can be shown in Algorithm 1 and Algorithm 2.

---

**Algorithm 1** Training Algorithm of DualDiff

**Input:** input graph $\mathcal{G}$, denoising networks $D_{\theta_l}$ and $D_{\theta_g}$, autoencoder $\mathcal{E}_\phi$ and $\mathcal{D}_\psi$, noise scheduler $\{\sigma_t\}$, diffusion steps $T$, training epochs $E$.
**Output:** learned $D_{\theta_l}$ and $D_{\theta_g}$.
1: Pretrain $\mathcal{E}_\phi$ and $\mathcal{D}_\psi$ by optimizing (2), and then fix the parameters.
2: $\boldsymbol{Z}_{l,0}, \boldsymbol{Z}_{g,0} \sim q_\phi(\cdot|\mathcal{G})$
3: **for** $i = 0$ to $E - 1$ **do**
4:     $t \sim \mathcal{U}(0, T)$
5:     $\boldsymbol{Z}_{l,t}, \boldsymbol{Z}_{g,t} \sim p_{\sigma_t}(\tilde{\boldsymbol{Z}}_l, \tilde{\boldsymbol{Z}}_g | \boldsymbol{Z}_{l,0}, \boldsymbol{Z}_{g,0})$
6:     Obtain $\boldsymbol{C}_l$ and $\boldsymbol{C}_g$ by (8).
7:     Update $D_{\theta_l}$ and $D_{\theta_g}$ by (11).
8: **end for**
9: **Return:** $D_{\theta_l}, D_{\theta_g}$

---

**Algorithm 2** Sampling Algorithm of DualDiff

**Input:** trained $D_{\theta_l}$ and $D_{\theta_g}$, trained decoder $\mathcal{D}_\psi$, noise scheduler $\{\sigma_t\}$, sampling steps $T$.
**Output:** generated graph $\hat{\mathcal{G}}$.
1: $\hat{\boldsymbol{Z}}_{l,T}, \hat{\boldsymbol{Z}}_{g,T} \sim$ prior
2: **for** $t = T - 1$ to $0$ **do**
3:     **if** $t \% (m + 1) == 0$ **then**
4:         $(\boldsymbol{C}_l, \boldsymbol{C}_g) = (\boldsymbol{0}, (\hat{\boldsymbol{Z}}_{l,0}, \hat{\boldsymbol{Z}}_{g,0}))$
5:     **else** $(\boldsymbol{C}_l, \boldsymbol{C}_g) = ((\hat{\boldsymbol{Z}}_{l,0}, \hat{\boldsymbol{Z}}_{g,0}), \boldsymbol{0})$
6:     $\hat{\boldsymbol{Z}}_{l,t} = \text{solver}(D_{\theta_l}(\hat{\boldsymbol{Z}}_{l,t+1}, \boldsymbol{C}_l, \sigma_{t+1}))$
7:     $\hat{\boldsymbol{Z}}_{g,t} = \text{solver}(D_{\theta_g}(\hat{\boldsymbol{Z}}_{g,t+1}, \boldsymbol{C}_g, \sigma_{t+1}))$
8: **end for**
9: $\hat{\mathcal{G}} = p_\psi(\hat{\boldsymbol{Z}}_{l,0}, \hat{\boldsymbol{Z}}_{g,0})$
10: **Return:** $\hat{\mathcal{G}}$

---

## 5 EXPERIMENT

Experimentally, we conduct comprehensive evaluations of our proposed DualDiff framework on two widely adopted graph generation tasks: generic graph generation and molecular generation.

### 5.1 EXPERIMENTAL SETUP

**Datasets.** Following previous methods, we leverage eight graph datasets to evaluate our proposed DualDiff extensively. For generic graph generation tasks, we include Ego-small, Community-small, Grid, Planar, and SBM datasets. For molecular graphs, we involve mainstream ZINC250k, QM9, and MOSES datasets. For a fair comparison, we follow the experimental and evaluation setting of (Liu et al., 2025; Jo et al., 2022; Vignac et al., 2022) with the same train/test split. More details and descriptions of datasets can be found in Appendix B.2.

**Baselines.** We compare DualDiff with competitive generative models. Specifically, for generic graph generation, we choose competitive GraphRNN (You et al., 2018), GDSS (Jo et al., 2022), DiGress (Vignac et al., 2022), GSDM (Luo et al., 2023), Graphusion (Yang et al., 2024), GruM (Jo et al., 2023), GBD (Liu et al., 2025), and GraphBFN (Song et al., 2025a). For molecular generation, we additionally select advanced MoLeR (Maziarz et al., 2021), GraphArm (Kong et al., 2023), MicCaM (Kong et al., 2023), MAGNet (Geng et al., 2023), , SwinGNN (Yan et al., 2023), LGD (Zhou et al., 2024), and EDM-SyCo (Ketata et al., 2025).

**Evaluation Metrics.** Following (You et al., 2018; Du et al., 2021; Jo et al., 2022), for generic graph generation, we use the maximum mean discrepancy (MMD) to compare the distributions of statistics between the same number of generated and test graphs, such as distributions of degree (Deg.), clustering coefficient (Clus.), the number of occurrences of orbits with 4 nodes (Orbit), and the eigenvalues of the graph Laplacian (Spec.). In addition, we report V.U.N. scores, representing the percentages of valid, unique, and novel graphs, to evaluate how well the model captures both

intrinsic characteristics and global graph properties. For molecular generation, we follow the metrics used in Guacamol (Brown et al., 2019), and report scores of KL divergence (KL), the Fréchet ChemNet Distance (FCD) (Preuer et al., 2018), Novelty, Uniqueness, and Validity.

Table 1: Comparison of advanced models on Planar and SBM datasets. More experiments on the Ego-Small, Community-small, and Grid datasets are included in Appendix C.1.

| Datasets | Planar | | | | | SBM | | | | |
|---|---|---|---|---|---|---|---|---|---|---|
| Metrics | Deg.↓ | Clus.↓ | Orbit↓ | Spec.↓ | V.U.N.↑ | Deg.↓ | Clus.↓ | Orbit↓ | Spec.↓ | V.U.N.↑ |
| Training set | 0.0002 | 0.0310 | 0.0005 | 0.0052 | 100.0 | 0.0008 | 0.0332 | 0.0255 | 0.0063 | 100.0 |
| GraphRNN | 0.0049 | 0.2779 | 1.2543 | 0.0459 | 0.0 | 0.0055 | 0.0584 | 0.0785 | 0.0065 | 5.0 |
| GDSS | 0.0041 | 0.2676 | 0.1720 | 0.0370 | 0.0 | 0.0212 | 0.0646 | 0.0894 | 0.0128 | 5.0 |
| DiGress | **0.0003** | 0.0372 | 0.0009 | 0.0106 | 75.0 | 0.0013 | 0.0498 | 0.0434 | 0.0400 | 74.0 |
| GBD | **0.0003** | 0.0353 | 0.0135 | 0.0059 | 92.5 | 0.0013 | 0.0493 | 0.0043 | 0.0047 | 75.0 |
| GruM | 0.0005 | 0.0353 | 0.0009 | 0.0062 | 90.0 | 0.0007 | 0.0492 | 0.0448 | 0.0050 | 85.0 |
| GraphBFN | 0.0005 | 0.0294 | **0.0002** | 0.0046 | 96.7 | 0.0005 | 0.0560 | 0.0370 | 0.0053 | **87.5** |
| **DualDiff** | **0.0003** | **0.0275** | **0.0002** | **0.0038** | **97.5** | **0.0004** | **0.0473** | **0.0365** | **0.0042** | 85.0 |

Table 2: Experiments on ZINC250K dataset. Following previous studies, we report the FCD and KL scores, where **higher values indicate better performance**. The results are reported from EDM-SyCo and the original papers. Methods that do not report the KL metric are denoted as N.A.

| | Method | FCD score (↑) | KL score (↑) | Novelty (↑) | Uniqueness (↑) | Validity (↑) |
|---|---|---|---|---|---|---|
| **Autoreg.** | GraphAF | 0.05 ± 0.00 | 0.67 ± 0.01 | 0.91 ± 0.01 | 0.91 ± 0.01 | **1.00 ± 0.00** |
| | **MoLeR** | **0.83 ± 0.00** | **0.97 ± 0.00** | 0.99 ± 0.00 | 0.99 ± 0.00 | **1.00 ± 0.00** |
| | GraphArm | 0.04 ± 0.00 | N.A. | **1.00 ± 0.00** | 0.99 ± 0.00 | 0.88 ± 0.00 |
| | MiCaM | 0.63 ± 0.02 | 0.94 ± 0.00 | 0.98 ± 0.00 | 0.98 ± 0.00 | **1.00 ± 0.00** |
| | MAGNet | 0.76 ± 0.00 | 0.95 ± 0.00 | 0.99 ± 0.00 | 0.99 ± 0.00 | **1.00 ± 0.00** |
| **One-shot** | GDSS | 0.10 ± 0.01 | N.A. | **1.00 ± 0.00** | **1.00 ± 0.00** | 0.97 ± 0.01 |
| | DiGress | 0.65 ± 0.00 | 0.91 ± 0.00 | 0.99 ± 0.00 | 0.99 ± 0.00 | 0.85 ± 0.01 |
| | GruM | 0.64 ± 0.01 | N.A. | **1.00 ± 0.00** | **1.00 ± 0.00** | **0.99 ± 0.00** |
| | SwinGNN | 0.67 ± 0.00 | N.A. | 0.96 ± 0.00 | **1.00 ± 0.00** | 0.91 ± 0.00 |
| | EDM-SyCo | 0.85 ± 0.01 | 0.96 ± 0.00 | **1.00 ± 0.00** | **1.00 ± 0.00** | 0.88 ± 0.01 |
| | **DualDiff** | **0.91 ± 0.02** | **0.98 ± 0.01** | **1.00 ± 0.00** | **1.00 ± 0.00** | 0.92 ± 0.02 |

## 5.2 GENERIC GRAPH GENERATION

From Table 1, we can observe that our proposed DualDiff achieves competitive performance across diverse metrics. Essentially, generic graphs demonstrate pronounced geometric properties. The topologically-enhanced global information can effectively capture the core features of the dataset, thereby providing valuable guidance for the generation of local information. We further validate our model's performance on the mainstream Community-small, Ego-small, and Grid datasets, as shown in Table 8, further illustrating the effectiveness and applicability of our model.

## 5.3 MOLECULAR GENERATION

We also evaluate DualDiff on molecular generation tasks. As shown in Table 2, DualDiff surpasses other advanced methods on the FCD and KL metrics, and exhibits competitive performance on the novelty and validity metrics. Fundamentally, molecular structures demonstrate dependencies across multiple scales, encompassing both atomic dependencies within motifs and interactions between distinct motifs. By equipping the generative model with both global and local awareness, DualDiff can substantially improve the quality of the generated molecules. Besides, most methods achieve novelty and uniqueness scores approaching 100%, which can be primarily attributed to the large molecular sizes. The resulting exponential growth in the number of possible molecular structures makes it difficult to sample molecules identical to those in the training and test sets.

Due to the page limits, we present the performance of our model on the QM9 dataset in Appendix C.2 and on the large-scale molecular dataset MOSES in Appendix C.4. In both cases, DualDiff

achieves competitive results compared to other advanced methods, further demonstrating the effectiveness of our approach. In addition to unconditional graph generation, DualDiff can also be applied to conditional generation tasks. Detailed experimental results are provided in Appendix C.5, where DualDiff likewise demonstrates strong conditional generation performance.

## 5.4 COMPREHENSIVE COMPARISON WITH HIERARCHICAL MODELS

Hierarchical graph generation methods Karami (2024); Bergmeister et al. (2024), which model complex graphs by decomposing them into multiple levels and progressively generating structures from coarse to fine, have achieved notable success in graph generation. In contrast to these autoregressive and coarse-to-fine modeling approaches, our dual conditioning mechanism enables dynamic interaction between global and local information. This allows for a more accurate capture of the joint distribution of global and local features compared to hierarchical methods. We provide a comparison of model performance between our method and hierarchical approaches, as shown in Table 3, further demonstrating the effectiveness of our approach.

Table 3: Comparison between advanced hierarchical models.

| Methods | Planar | | | | SBM | | | |
|---|---|---|---|---|---|---|---|---|
| Metrics | Deg. | Clus. | Orbit | Spec. | Deg. | Clus. | Orbit | Avg. |
| PPGN (Bergmeister et al., 2024) | **0.0003** | **0.0245** | 0.0006 | 0.0104 | 0.0119 | 0.0517 | 0.0669 | 0.0067 |
| HiGen (Karami, 2024) | 0.0012 | 0.0435 | 0.0234 | **0.0025** | 0.0017 | 0.0503 | 0.0604 | 0.0068 |
| DualDiff (Ours) | **0.0003** | 0.0275 | **0.0002** | 0.0038 | **0.0004** | **0.0473** | **0.0365** | **0.0042** |

## 5.5 EXPERIMENTS ON GENERATING 3D MOLECULES

In this subsection, we will evaluate the effectiveness of our approach in 3D molecular tasks. Following GEOLDM (Xu et al., 2023), we replace the lightweight MLP decoder with an EGNN to ensure that both latent embedding and coordinates are simultaneously generated. The experimental results on the QM9 dataset are shown in Table 4. Our method achieves competitive performance compared to the advanced GEOLDM and EQUIFM (Song et al., 2023), which further illustrates the effectiveness of our methods on generating 3D molecules.

Table 4: Experiments of 3D molecular generation on the QM9 Dataset.

| Methods | Atom Sta (%) | Mol Sta (%) | Valid & Unique (%) |
|---|---|---|---|
| Data | 99.0 | 95.2 | 97.7 |
| GEOLDM | 98.9 | 89.4 | 92.7 |
| EQUIFM | 98.9 | 88.3 | 93.5 |
| DualDiff | 98.9 | 88.7 | 99.3 |

## 5.6 MODEL ANALYSIS

**Ablation study.** We conduct an ablation study on the proposed dual conditioning mechanism, as shown in Table 5. Specifically, we compare the model performance of DualDiff without any conditioning, with self-conditioning, with only global-to-local conditioning, and with full dual conditioning on the ZINC250k dataset.

The results demonstrate that dual conditioning can substantially improve the quality of the generated graphs and significantly outperforms the conventional self-conditioning approach, underscoring the importance of the interaction between global and local topological information.

Table 5: Ablation study of dual conditioning.

| Method | FCD ($\uparrow$) | KL ($\uparrow$) |
|---|---|---|
| DualDiff (w/o any cond.) | $0.65 \pm 0.01$ | $0.82 \pm 0.02$ |
| DualDiff (w self cond.) | $0.72 \pm 0.01$ | $0.89 \pm 0.02$ |
| DualDiff ($\boldsymbol{Z}_l \rightarrow \boldsymbol{Z}_g$) | $0.75 \pm 0.03$ | $0.95 \pm 0.02$ |
| DualDiff ($\boldsymbol{Z}_g \rightarrow \boldsymbol{Z}_l$) | $0.83 \pm 0.01$ | $0.95 \pm 0.01$ |
| DualDiff ($\boldsymbol{Z}_g \leftrightarrow \boldsymbol{Z}_l$) | $\mathbf{0.91 \pm 0.02}$ | $\mathbf{0.98 \pm 0.01}$ |

**Parameter Analysis.** We conduct an extensive evaluation of model performance under varying key parameters, including the frequency $p$ of process (i) during training, $m$ steps used in Section 4.3, and the number of clusters $K$. Detailed

results on the ZINC250k dataset are presented in Figure 3. The left panel indicates that selecting a moderate value for $p$ facilitates effective interaction between global and local information. The middle panel shows that increasing $m$ during sampling enables the model to focus more on complex local details, leading to enhanced performance. Finally, the right panel highlights the significance of an appropriate cluster number $K$: a smaller $K$ not only reduces computational overhead but also preserves essential global information. Besides, we also include the detailed ablation studies of different clustering methods, and cluster numbers in Appendix C.6.

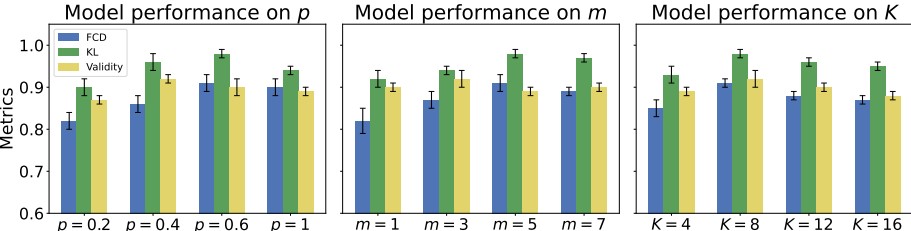

Figure 3: Experiments of model parameters on ZINC250k dataset.

**Model Efficiency.** Apart from the advanced generation performance, our model also demonstrates high efficiency. As shown in Table 6, DualDiff achieves competitive results using only approximately 200 steps. We further evaluate the actual time to generate 10,000 molecular graphs on the QM9 and ZINC250k datasets, as reported in Table 7, where "w/o dual." refers to the time without dual conditioning. The results show that the dual conditioning mechanism introduces acceptable computational overhead, and our model consistently maintains high efficiency. Additionally, we provide further explanations and experiments regarding the efficiency of DualDiff in Appendix C.7.

Table 6: Model performance at different diffusion steps on Community-small and Ego-small datasets.

Table 7: Experimental evaluation of the model's sampling efficiency.

| Dataset | Models | 50 | 100 | 200 | 500 | 1000 |
|---------|--------|------|------|------|------|------|
| Comm. | GDSS | 0.110 | 0.104 | 0.072 | 0.061 | 0.046 |
| | GSDM | 0.109 | 0.079 | 0.021 | 0.011 | 0.009 |
| | DualDiff | 0.098 | 0.043 | 0.009 | 0.007 | 0.006 |
| Ego. | GDSS | 0.040 | 0.031 | 0.023 | 0.021 | 0.017 |
| | GSDM | 0.044 | 0.026 | 0.024 | 0.019 | 0.016 |
| | DualDiff | 0.032 | 0.012 | 0.005 | 0.005 | 0.004 |

| Dataset | Model | Sampling (s) |
|---------|-------|-------------|
| QM9 | GDSS | 109 |
| | DualDiff | 41 |
| | - w/o dual. | 32 |
| ZINC250k | GDSS | 1870 |
| | DualDiff | 223 |
| | - w/o dual. | 169 |

# 6    LIMITATIONS AND FUTURE DIRECTIONS

The proposed DualDiff is currently primarily designed for generating generic and simple molecular graphs. However, as a general paradigm, DualDiff can effectively model local and global information simultaneously, and thus can be broadly transferred to other data with hierarchical characteristics, such as proteins, peptides, and even circuit design. We consider these extensions as promising directions for future work and will explore the applicability of DualDiff to diverse domains.

# 7    CONCLUSION

In this paper, we design a unified global and local topology-aware generative model, i.e., DualDiff. DualDiff employs a two-branch diffusion framework and dual conditioning mechanism, which can effectively model the joint distribution between global and local information, thereby naturally capturing the complex and entangled multi-scale relationships inherent in graphs. Empirical results demonstrate that our proposed model significantly enhances the quality of generated graphs while further ensuring training stability and sampling efficiency.

## ACKNOWLEDGMENTS

The research work described in this paper was conducted in the JC STEM Lab of Machine Learning and Symbolic Reasoning, funded by The Hong Kong Jockey Club Charities Trust.

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

# A   MODEL DETAILS

## A.1   AUTOENCODER AND LATENT SPACE

**Encoder.**   In this subsection, we present the details of the encoder. For generic graphs, we employ message-passing graph neural networks (MPNNs) (Gilmer et al., 2017) to implement $\mathcal{E}_\phi$. The latent representation $\boldsymbol{Z}_v$ for each node $v$ is computed via layer-wise message passing as follows:

$$\boldsymbol{Z}_v^{(k)} = \text{UPDATE}^{(k)}\Big(\boldsymbol{Z}_v^{(k-1)}, \text{AGG}^{(k)}\big(\boldsymbol{Z}_u^{(k-1)} : u \in \mathcal{N}(v)\big)\Big), \tag{12}$$

where $k$ denotes the $k$-th layer, UPDATE and AGG correspond to the update and aggregation functions, and $\mathcal{N}(v)$ denotes the set of neighbors of node $v$. For molecular graphs, atomic coordinates provide informative representations of inter-node relationships. Therefore, we incorporate the equivariant graph neural network (EGNN) to implement $\mathcal{E}_\phi$, enabling the mapping $\mathcal{G} \to \boldsymbol{Z}$ and producing coordinate-enhanced node embeddings, i.e., $\boldsymbol{Z} = (\boldsymbol{Z}^{(h)}, \boldsymbol{Z}^{(x)})$.

**Decoder.**   To reconstruct graph-structured data from the latent embedding $\boldsymbol{Z}_l$ and $\boldsymbol{Z}_g$, the decoder first combines both local and global information. Then, it leverages lightweight MLPs to separately reconstruct the graph features $\boldsymbol{H}$ and the adjacency matrix $\boldsymbol{A}$. Specifically, the decoder first utilizes a FiLM layer to integrate global and local information, obtaining global context-enhanced node embeddings, as follows:

$$\gamma, \beta = \text{Linear}(\boldsymbol{S}_g \boldsymbol{Z}_g), \tilde{\boldsymbol{Z}}_l = \gamma \odot \boldsymbol{Z}_l + \beta, \tag{13}$$

where $\boldsymbol{S}_g \in \mathbb{R}^{N \times K}$ is the affiliation matrix, which can be obtained based on the similarity between $\boldsymbol{Z}_g$ and $\boldsymbol{Z}_l$. Then, our model employs two separate lightweight MLPs to recover graph-structured data, i.e., the node features and adjacency matrix. Formally, we have:

$$\hat{\boldsymbol{X}} = \text{MLP}_{\text{node}}(\tilde{\boldsymbol{Z}}_l), \tag{14}$$

$$\hat{\boldsymbol{A}} = \sigma(\text{MLP}_{\text{adj}}(\tilde{\boldsymbol{Z}}_l W \tilde{\boldsymbol{Z}}_l^T)), \tag{15}$$

where $\gamma$ and $\beta$ denote the scale and shift parameters induced by the global cluster embedding, $\sigma$ is Sigmoid activation function, $\hat{\boldsymbol{X}}$ and $\hat{\boldsymbol{A}}$ represent the reconstructed node and adjacency matrix. For molecular graphs, we can further leverage the coordinate information to facilitate more accurate adjacency matrix reconstruction, which can be formulated as follows:

$$\hat{\boldsymbol{A}}_{ij} = \text{MLP}_{\text{adj}}\Big(\text{Concat}\Big[\boldsymbol{Z}_i^{(h)}, \boldsymbol{Z}_j^{(h)}, \|\boldsymbol{Z}_i^{(x)} - \boldsymbol{Z}_j^{(x)}\|_2^2\Big]\Big). \tag{16}$$

**Latent Space.**   Unlike discrete graph generation paradigms, we follow the LGD paradigm by mapping the graph space into a continuous latent space. This approach allows us to fully leverage informative node features to enhance the quality of the generated graphs. Besides, traditional discrete graph diffusion frameworks require explicit diffusion over the entire adjacency matrix, resulting in $O(N^2)$ complexity. In contrast, node embeddings in the latent space can implicitly model the relationships among nodes. Our latent diffusion approach operates at the node level, with a diffusion space complexity of $O(N)$, which significantly enhances computational efficiency.

## A.2   GLOBAL INFORMATION EXTRACTION

We leverage powerful graph clustering methods to extract topology-aware global information. Specifically, for generic graphs without explicit coordinate information, we first perform spectral decomposition on the graph Laplacian matrix, $\boldsymbol{L} = \boldsymbol{D} - \boldsymbol{A}$, yielding $\boldsymbol{L} = \boldsymbol{U}\boldsymbol{\Lambda}\boldsymbol{U}^T$. We then select the leading $K$ eigenvectors of $\boldsymbol{U}$ and construct $\boldsymbol{U}' = [u_1, u_2, \ldots, u_K] \in \mathbb{R}^{N \times K}$. The K-means algorithm is subsequently applied to the rows of $\boldsymbol{U}'$ to obtain the final clustering results, denoted as $\boldsymbol{S}_g$. Additionally, we incorporate the eigenvectors $\boldsymbol{U}$ into the node embeddings to inject meaningful topological information, which can largely enhance the reconstruction ability of the autoencoder.

For molecular graphs, we can utilize atomic coordinates to perform clustering by directly applying K-means to the atom-level coordinate representations, $\boldsymbol{Z}^{(x)}$. The resulting cluster centers can be regarded as the centers of molecular fragments. This global structural information can provide meaningful guidance, steering molecular generation towards more interpretable and chemically relevant structures. After clustering and obtaining the affiliation matrix $\boldsymbol{S}_g$, we aggregate the embeddings of nodes belonging to the same cluster and ultimately obtain $\boldsymbol{Z}_g \in \mathbb{R}^{K \times d}$.

### A.3 DENOISING NETWORK

In this subsection, we include a detailed description of the architectures used in the denoising networks $D_{\theta_l}$ and $D_{\theta_g}$. For general graph generation tasks, both $D_{\theta_l}$ and $D_{\theta_g}$ are implemented using the Graph Transformer, which enables global all-pair message passing by aggregating information from all node embeddings to update each node's representation. For molecular generation tasks, we adopt EGNN for $D_{\theta_l}$ and $D_{\theta_g}$ to ensure the equivariance of the generation process.

### A.4 DUAL CONDITIONING MECHANISM

In this subsection, we include more details of the proposed dual conditioning mechanism. During the two processes of the dual conditioning mechanism, we can leverage the node-to-cluster similarity to achieve dynamic global and local information exchange without explicit clustering. Specifically, we first obtain the estimated outputs from the previous step: $\hat{Z}_{l,0} = D_{\theta_l}(Z_{l,t}, C_l)$ and $\hat{Z}_{g,0} = D_{\theta_g}(Z_{g,t}, C_g)$. This step is analogous to self-conditioning, where the output from the previous step is used as a condition for the current step. Then we can calculate the affiliation matrix $S_g \in \mathbb{R}^{N \times K}$ based on $\hat{Z}_{l,0}$ and $\hat{Z}_{g,0}$, where $S_g^{i,j} = 1$, if $j = \arg\max_j \text{sim}(\hat{Z}_{l,0}^i, \hat{Z}_{g,0}^j)$ and $S_g^{i,j} = 0$, otherwise. During process (i), we assign each node to its corresponding cluster according to $S_g$, and introduce the condition based on (9). During process (ii) in (10), we utilize message passing and average pooling operations to extract the initial global condition $\boldsymbol{C}$, where $\boldsymbol{C}^j = \frac{\sum_{i=1}^{N} S_g^{ij} \cdot \hat{Z}_{l,0}^i}{\sum_{i=1}^{N} S_g^{ij}}$.

## B EXPERIMENTAL SETTING

### B.1 METRICS

In Table 2, we report the scores of the FCD and KL metrics following (Brown et al., 2019). Specifically, the detailed calculation methods are as follows:

**KL score**. For both generated and reference molecules, we calculate the following descriptors: BertzCT, MolLogP, MolWt, TPSA, NumHAcceptors, NumHDonors, NumRotatableBonds, NumAliphaticRings, NumAromaticRings, and the ECFP4 fingerprint-based similarity to the nearest neighbor. The KL divergence $D_{KL,i}$ is computed for each descriptor to measure the difference between the two molecular sets. These divergences are then combined to produce a final normalized score, given by $\frac{1}{k} \sum_{i=1}^{k} \exp(-D_{KL,i})$.

**FCD score**. FCD is evaluated using the hidden representations from ChemNet, a neural network trained to predict biological activities. Specifically, we extract the means and covariances of the last hidden activations in ChemNet for both the generated and reference molecules. The Fréchet distance between these two distributions is then calculated. To obtain a final score between 0 and 1, this distance is normalized using $\exp(-0.2 \cdot FCD)$.

### B.2 DATASETS

Following previous methods, we leverage eight graph datasets to evaluate our proposed DualDiff extensively. The datasets include: (1) Ego-small: 200 small ego graphs collected from Citeseer graph datasets (Sen et al., 2008); (2) Community-small: 100 random graphs generated by (ERDdS & R&wi, 1959); (3) Grid: 100 standard 2D grid graphs; (4) Planar: 200 synthetic planar graphs with $N = 64$ nodes; (5) SBM: 200 stochastic block model graphs with 2–5 communities, where $44 \leq N \leq 187$ and each community contains 20–40 nodes; (6) ZINC250k: 249,455 molecular graphs with 9 node types (Irwin et al., 2012); (7) QM9: 133,885 molecular graphs with 4 node types (Ramakrishnan et al., 2014); (8) MOSES: large-scale molecular dataset (Polykovskiy et al., 2020).

### B.3 TRAINING DETAILS

All experiments are conducted on NVIDIA 4090 GPUs. We employ the Adam optimizer with a learning rate selected from $\{1 \times 10^{-3}, 5 \times 10^{-4}, 1 \times 10^{-4}\}$. The sampling steps are fixed at 256. Our parameters follow the settings in the Elucidated Diffusion Models (EDM), with $\sigma_{min} = 0.002$,

$\sigma_{max} = 80$, and $\rho = 7$. The hidden dimensions of $\boldsymbol{Z}_l$ and $\boldsymbol{Z}_g$ are set to $d \in \{32, 64, 128\}$. Besides, we choose $2^{nd}$ order Heun as the ODE solver. To ensure equivalence in molecular generation tasks, we also subtract the center of mass from the noise added to the coordinates at each step. Each experiment is repeated three times, and we report the average results.

For molecular generation, the encoder utilizes a 2-layer EGNN architecture, while a 2-layer MPNN is used for generic graph generation. The decoder is composed of two separate MLPs, responsible for reconstructing edge and node features, respectively. For molecular tasks, $D_{\theta_l}$ and $D_{\theta_g}$ are both implemented by 4-layer EGNNs; for generic graphs, we employ 4-layer Graph Transformers.

## C  ADDITIONAL EXPERIMENTAL RESULTS

### C.1  EXPERIMENTS ON OTHER GENERIC GRAPH DATASETS

In this subsection, we further evaluate the model's performance on the Ego-small, Community-small, and Grid datasets, as shown in Table 8. From Table 8, it can be clearly observed that our proposed method achieves competitive performance across four widely used generic graph generation datasets. For Ego-small and Community-small datasets that display clear global topological patterns, our proposed DualDiff can effectively capture both global and local information, exhibiting advanced generation performance. Besides, on larger datasets such as Grid datasets, DualDiff maintains strong performance, further demonstrating its scalability and robustness. Essentially, generic graphs typically exhibit prominent global information. Incorporating global awareness during the generation process can effectively guide the generation process toward a meaningful direction, accelerate training convergence, and improve the final quality of the generated results. To further demonstrate the effectiveness of our method, we also include visualization results of graphs generated by DualDiff in Appendix I.

Table 8: Experiments on generic graph generation. We report MMD distances of different statistics between the test set and the generated graphs. Avg. means the average results of three metrics.

| Methods | Ego-small Real, $4 \leq |V| \leq 18$ | | | | Community-small Synthetic, $12 \leq |V| \leq 20$ | | | | Grid Synthetic, $100 \leq |V| \leq 400$ | | | |
|---|---|---|---|---|---|---|---|---|---|---|---|---|
| | Deg. | Clus. | Orbit | Avg. | Deg. | Clus. | Orbit | Avg. | Deg. | Clus. | Orbit | Avg. |
| GraphRNN | 0.090 | 0.220 | 0.003 | 0.104 | 0.080 | 0.120 | 0.040 | 0.080 | 0.064 | 0.043 | 0.021 | 0.043 |
| GraphAF (Shi et al., 2020) | 0.034 | 0.110 | **0.001** | 0.047 | 0.181 | 0.205 | 0.023 | 0.133 | – | – | – | – |
| GraphDF (Luo et al., 2021) | 0.041 | 0.135 | 0.010 | 0.060 | 0.064 | 0.125 | 0.037 | 0.070 | – | – | – | – |
| GDSS | 0.021 | 0.024 | 0.007 | 0.017 | 0.045 | 0.086 | 0.007 | 0.046 | 0.111 | 0.005 | 0.070 | 0.062 |
| DiGress | 0.025 | 0.021 | 0.008 | 0.018 | 0.051 | 0.083 | 0.009 | 0.048 | 0.189 | 0.003 | 0.075 | 0.089 |
| GSDM | 0.021 | 0.022 | 0.007 | 0.016 | 0.011 | 0.015 | **0.001** | 0.009 | 0.002 | **0.0** | **0.0** | 0.0007 |
| GraphBFN | 0.011 | 0.014 | 0.002 | 0.009 | **0.002** | 0.060 | 0.002 | 0.021 | 0.045 | 0.011 | 0.040 | 0.032 |
| **DualDiff (Ours)** | **0.007** | **0.008** | **0.001** | **0.005** | 0.007 | **0.012** | 0.002 | **0.007** | **0.001** | **0.0** | **0.0** | **0.0003** |

### C.2  EXPERIMENTS ON QM9 DATASET

To comprehensively evaluate the effectiveness of our model on molecular generation tasks, we also followed the experimental settings of GDSS. In addition to the FCD metric, we evaluated the neighborhood subgraph pairwise distance kernel (NSPDK) MMD (Costa & De Grave, 2010), which measures the graph kernel distance between subgraph structures and node features. The results, as shown in Table 9, demonstrate that our proposed DualDiff model outperforms other state-of-the-art methods, further highlighting its competitive performance in molecular generation tasks.

### C.3  EXPERIMENTS ON PROTEINS DATASET

To further validate the effectiveness of our methods on large generic graphs, we compare DualDiff with other advanced generative methods, as shown in Table 10. The results further illustrate the effectiveness and applicability of our methods.

Table 9: Experiments of different generative methods on QM9 dataset.

| Method | FCD ($\downarrow$) | NSPKD ($\downarrow$) | Unique ($\uparrow$) | Valid w/o cor. ($\uparrow$) |
|---|---|---|---|---|
| GDSS | 2.900 | 0.003 | 0.95 | 0.96 |
| GSDM | 2.650 | 0.003 | 0.97 | **0.99** |
| Graphusion | 2.593 | 0.004 | **0.99** | 0.97 |
| DiGress | 0.095 | 0.0003 | 0.98 | 0.98 |
| GruM | 0.108 | 0.0002 | 0.97 | **0.99** |
| LGD(Zhou et al., 2024) | 0.100 | 0.0002 | 0.98 | 0.98 |
| GBD | 0.093 | 0.0002 | 0.96 | **0.99** |
| GraphBFN | 0.101 | 0.0002 | 0.98 | 0.99 |
| **DualDiff (Ours)** | **0.092** | **0.0001** | 0.97 | 0.98 |

Table 10: Experiments of diffrent methods on the Proteins Dataset.

| Methods | Deg. | Clust. | Orbit | Spec. |
|---|---|---|---|---|
| GraphRNN | 0.004 | 0.1475 | 0.5851 | 0.0152 |
| DiGress | 0.0041 | 0.0489 | 0.1286 | 0.0018 |
| GDSS | 0.0861 | 0.5111 | 0.732 | 0.0748 |
| GruM | 0.0019 | 0.0660 | 0.0345 | 0.0030 |
| DualDiff | 0.0022 | 0.0309 | 0.0067 | 0.0030 |

## C.4 EXPERIMENTS ON LARGE-SCALE GENERATION

To further illustrate the effectiveness of our methods in large-scale generation scenarios, we also evaluate our methods on the MOSES datasets (Polykovskiy et al., 2020). From Table 11, our proposed DualDiff achieves comparable performance with advanced DiGress and LGD in terms of validity and novelty metrics. Besides, DualDiff can further bridge the gap between diffusion-based graph generation models and other traditional methods, illustrating its effectiveness.

Table 11: Experiments of large-scale generation on MOSES dataset.

| Model | Class | Valid | Unique | Novelty | FCD ($\downarrow$) |
|---|---|---|---|---|---|
| JT-VAE (Jin et al., 2018) | Fragment | 100 | 100 | 99.9 | 1.00 |
| GraphINVENT (Mercado et al., 2021) | Autoreg. | 96.4 | 99.8 | - | 1.22 |
| ConGress (Vignac et al., 2022) | One-shot | 83.4 | 99.9 | 96.4 | 1.48 |
| DiGress (Vignac et al., 2022) | One-shot | 85.7 | 100 | 95.0 | 1.19 |
| LGD (Zhou et al., 2024) | One-shot | 97.4 | 100 | 95.9 | 1.42 |
| GraphBFN (Song et al., 2025b) | One-shot | 88.5 | 99.8 | 89.0 | 1.07 |
| **DualDiff (Ours)** | One-shot | 96.8 | 100 | 96.2 | 1.16 |

## C.5 EXPERIMENTS ON CONDITIONAL GENERATION

We also extend our proposed DualDiff to conditional generation scenarios. Following the settings in GEOLDM and LGD, we evaluate the controllable molecule generation given the desired property value $s$. We use a domain-specific condition encoder $\tau$ to obtain the embedding of the property conditions $\tau(s)$. Then, the autoencoder and denoising networks take the condition embeddings as additional inputs, shifting the latent representations towards the data distribution aligned with the condition. Specifically, following GEOLDM, we split QM9 dataset into two parts, each including 50k molecules. Then, we train a property prediction network and our diffusion model on these two halves, respectively. For evaluation, we generate samples given the property $s$, and leverage the property prediction network to predict the property $\hat{s}$ of the generated samples. Finally, we report the Mean Absolute Error (MAE) between $s$ and $\hat{s}$ to measure the controllable performance of DualDiff. In this experiment, we consider six important chemical properties: polarizability $\alpha$, orbital energies $\varepsilon_{\text{HOMO}}$, $\varepsilon_{\text{LUMO}}$ and their gap $\Delta\varepsilon$, dipole moment $\mu$, and heat capacity $C_v$. From Table 12, DualDiff demonstrates competitive controllable generation performance with advanced LGD and GEOLDM, further illustrating its effectiveness on conditional generation tasks.

Table 12: Conditional generation results on QM9 (MAE $\downarrow$)

| Model | $\mu$ | $\alpha$ | $\epsilon_{\text{HOMO}}$ | $\epsilon_{\text{LUMO}}$ | $\Delta\epsilon$ | $c_v$ |
|---|---|---|---|---|---|---|
| Random | 1.616 | 9.01 | 645 | 1457 | 1470 | 6.857 |
| GeoLDM | 1.108 | 2.37 | 340 | 522 | 587 | 1.025 |
| LGD | 0.879 | 2.43 | 313 | 641 | 586 | 1.002 |
| **DualDiff (Ours)** | 0.834 | 2.35 | 322 | 587 | 582 | 0.987 |

## C.6 ABLATION STUDY ON DIFFERENT CLUSTERING METHODS

To further validate the effectiveness of our model with different clustering algorithms, we evaluate the impact of various clustering methods on the QM9 dataset, as shown in Table 13. The results demonstrate that our approach remains effective across different clustering strategies.

Additionally, we discuss the influence of the number of clusters on model performance. We find that adopting a moderate number of clusters achieves a good balance between efficiency and accuracy, as shown in Table 14, which underscores the importance of selecting an appropriate cluster number.

Table 13: Experiments of different clustering methods.

| Clustering Method | FCD ($\downarrow$) | NSPKD ($\downarrow$) | Unique ($\uparrow$) | Valid w/o cor. ($\uparrow$) |
|---|---|---|---|---|
| Spectral Clustering | 0.094 | 0.0004 | 0.97 | 0.98 |
| GMM | 0.095 | 0.0006 | 0.96 | 0.97 |
| Louvain Algorithm | 0.100 | 0.0003 | 0.96 | 0.97 |
| K-means (Current) | 0.092 | 0.0001 | 0.95 | 0.98 |

Table 14: Experiments on the number of clusters.

| Methods Metrics | Enzymes | | | | Planar | | | |
|---|---|---|---|---|---|---|---|---|
| | Deg. | Clus. | Orbit | Avg. | Deg. | Clus. | Orbit | Avg. |
| 4 | 0.015 | 0.109 | 0.009 | 0.0443 | 0.0008 | 0.0415 | 0.0009 | 0.0104 |
| 6 | 0.013 | 0.111 | 0.010 | 0.0447 | **0.0003** | 0.0275 | **0.0002** | **0.0038** |
| 8 | **0.010** | **0.083** | **0.007** | **0.0333** | 0.0004 | **0.0271** | 0.0003 | 0.0064 |
| 10 | 0.012 | 0.087 | 0.007 | 0.0353 | **0.0003** | 0.0291 | **0.0002** | 0.0056 |

## C.7 FURTHER EXPLANATION OF DUAL CONDITIONING MECHANISM

To further illustrate the effectiveness of the dual conditioning mechanism, we compare the model performance with and without the dual conditioning mechanism under varying diffusion steps. The detailed results are shown in Table 15. We can observe that introducing the dual conditioning mechanism can achieve strong performance even with a very small number of steps, further highlighting the importance of information exchange between global and local representations.

## C.8 EXPLANATION OF THE MODEL EFFICIENCY

To further validate the overhead of the additional dual conditioning introduced in DualDiff, we conduct analyses from two perspectives: complexity analysis and empirical evaluation of practical sampling time and memory consumption.

## D EXPLANATION OF THE CLUSTERING METHODS

In the global information extraction, we perform clustering in either the 3D geometric space or the topological space of the graph to obtain topologically-based classes. By combining these geometrically meaningful clusters with node embeddings, we introduce a topological bias, which facilitates the extraction of effective global information. In general, the geometry or topology of the graph serves as a powerful, informative, and critical indicator, further ensuring the effectiveness of the global information extraction.

Table 15: Model performance of different conditioning methods at distinct diffusion steps.

| Dataset | Models | 50 | 100 | 200 | 500 | 1000 |
|---------|--------|-----|-----|-----|-----|------|
| Comm. | DualDiff (w/o any cond.) | 0.099 | 0.078 | 0.032 | 0.025 | 0.024 |
| | DualDiff (w. self cond.) | 0.100 | 0.064 | 0.018 | 0.016 | 0.016 |
| | DualDiff | 0.098 | 0.043 | 0.009 | 0.007 | 0.006 |

To further validate the effectiveness of the clustering methods used, we compare them with advanced learnable deep clustering methods. Learnable deep graph clustering or pooling methods extract global information based on the similarity of node embeddings, typically leveraging K-means clustering or pooling to obtain global representations. The key difference between our approach and these methods is that we extract global information in a training-free manner by utilizing geometry-based clusters, whereas deep clustering methods depend on the similarity of latent embeddings and require additional training time and computational resources. We provide a comparison between deep clustering and our approach, as shown below. Notably, K-means and spectral clustering achieve performance comparable to deep clustering, further demonstrating the effectiveness of topological bias in capturing global information.

## E  EXPLANATION OF THE SAMPLING PROCESS

### E.1  THEORETICAL INSIGHTS OF THE DUAL CONDITIONING

To model the high-dimensional and intractable joint probability distribution $P(Z_l, Z_g)$, diffusion-based methods often use the target $Z_{l,0}$ and $Z_{g,0}$ as conditions to model the conditional probability path $p_t$. This can be factorized into the following three equivalent forms:

$$p_t(Z_l, Z_g) = \int_{Z_{l,0}, Z_{g,0} \sim \text{data}} p_t(Z_l, Z_g | Z_{l,0}, Z_{g,0}) dZ_{l,0} dZ_{g,0} \tag{17}$$

$$= \int_{Z_{l,0}, Z_{g,0} \sim \text{data}} p_t(Z_l | Z_{l,0}) p_t(Z_g | Z_l, Z_{g,0}) dZ_{l,0} dZ_{g,0} \tag{18}$$

$$= \int_{Z_{l,0}, Z_{g,0} \sim \text{data}} p_t(Z_l | Z_g, Z_{l,0}) p_t(Z_g | Z_{g,0}) dZ_{l,0} dZ_{g,0} \tag{19}$$

These three factorizations provide alternative ways to model the joint conditional probability path $p_t$: 1) simultaneously incorporating both global and local conditions; 2) incorporating local information into the generation of global information; incorporating global information into the generation of local information.

Furthermore, in diffusion models, $p_t$ is typically instantiated as a Gaussian path. For example, in the decomposition $p_t(Z_l | Z_{l,0}), p_t(Z_g | Z_l, Z_{g,0}), p_t(Z_l | Z_{l,0})$ can be modeled as $\mathcal{N}(Z_l; \mu_t(Z_{l,0}), \sigma_t(Z_{l,0}))$, corresponding to a standard diffusion process without additional conditions, i.e., $C_l = 0$ in process (ii). Besides, $p_t(Z_g | Z_l, Z_{g,0}) = \mathcal{N}(Z_g; \mu_t(Z_l, Z_{g,0}), \sigma_t(Z_l, Z_{g,0}))$, which can be seen as a conditional diffusion process with $Z_l$ as the condition. By further integrating self-conditioning, we obtain $C_g = (\hat{Z}_{l,0}, \hat{Z}_{g,0})$. Similarly, in process (i), $(C_l, C_g) = (((\hat{Z}_{l,0}, \hat{Z}_{g,0}), 0))$ corresponds to modeling $p_t(Z_l | Z_g, Z_{l,0}) p_t(Z_g | Z_{g,0})$.

### E.2  EFFECTIVENESS OF THE ALTERNATING MECHANISM

During the sampling phase, we take inspiration from the server-client communication paradigm and employ a strategy that alternates $m$ steps of process (i) with a single step of process (ii). In this section, we provide a more detailed explanation of this approach. As illustrated in Figure 4, process (i) incorporates global information into the generation of local details, analogous to information flow from the server to the client for local updates. Conversely, during process (ii), local information is aggregated to improve the prediction of the global representation, which can be regarded as communication from the client to the server for global aggregation. To ensure a more stable

sampling process, we perform $m$ local updates for each global aggregation, which enhances overall performance and facilitates the sampling procedure.

In addition, this asymmetric alternating design is primarily intended to ensure stability during the sampling process and to promote a better match between global and local information. Essentially, the learning difficulty of global and local information is different. Generally speaking, learning local details is more complex than capturing global information. The asymmetric paradigm allows the sampling process to focus more on integrating global information into local details. In contrast, a symmetric design would introduce both global and local information into the learning process simultaneously during sampling. However, the quality of the generated local information may be significantly lower than that of global information, which may provide incorrect guidance to the global process and cause instability. To further validate the effectiveness of this asymmetric design, we conducted ablation studies on the asymmetry design, as shown in Table 16, which demonstrate the rationality of the asymmetric alternating strategy.

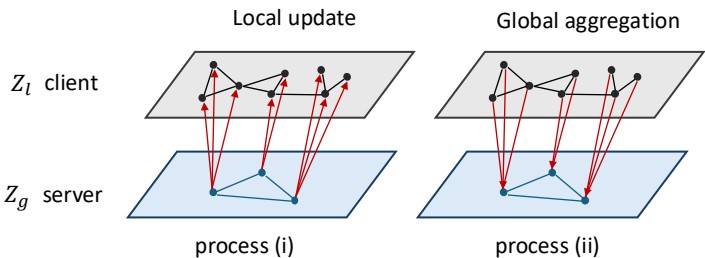

Figure 4: Explanation of the sampling process.

Table 16: Comparison of asymmetric and symmetric methods

| **Datasets** | Enzymes | | | Planar | | |
|---|---|---|---|---|---|---|
| asymmetric | **0.010** | **0.083** | **0.007** | **0.0003** | **0.0275** | **0.0002** |
| symmetric | 0.018 | 0.107 | 0.009 | 0.0009 | 0.0425 | 0.0004 |

With respect to convergence, the global information provides a "big picture" for the generation of local details, facilitating convergence and constraining the exploration space for local details. We further compare the performance of simultaneous and alternating methods under the same number of sampling steps. The results show that our approach can achieve superior performance with a much smaller number of steps.

## F  ETHIC STATEMENT

This work focuses on the development of novel algorithms for graph generation, with the primary goal of advancing research in machine learning. Our methods are intended to facilitate the design and discovery of new molecular structures and graph-based data representations, which may be beneficial for applications in materials science, drug discovery, and related fields. We believe that our work does not introduce negative social or ethical impacts.

## G  REPRODUCIBILITY STATEMENT

In this paper, we provide detailed model configurations, as presented in Appendix A. Besides, we also provide comprehensive experimental details, as shown in Appendix B. Additionally, the datasets used in the paper are all publicly available, ensuring consistent and reproducible evaluation results. The code will be available at https://github.com/Xyhi/DualDiff.

## H    LLM USAGE

Large language models (LLMs) were utilized to support the writing and editing process of this manuscript. In particular, we employed an LLM to help refine the language, enhance readability, and ensure clarity throughout various sections of the paper. It is important to note that the LLM was not involved in the ideation, research methodology, or experimental design. The authors take full responsibility for the content of the manuscript, including any text generated or polished by the LLM. We have ensured that the LLM-generated text adheres to ethical guidelines and does not contribute to plagiarism or scientific misconduct.

## I    VISUALIZATION

To provide a more intuitive demonstration of the effectiveness of our method, we present visualization results of DualDiff on datasets including Ego-small, Community-small, QM9, and ZINC250k. As shown in Figure 5, Figure 6, Figure 7 and Figure 8, the graphs generated by our method successfully capture the distinctive characteristics of each dataset, further highlighting the strong effectiveness and practical utility of our approach.

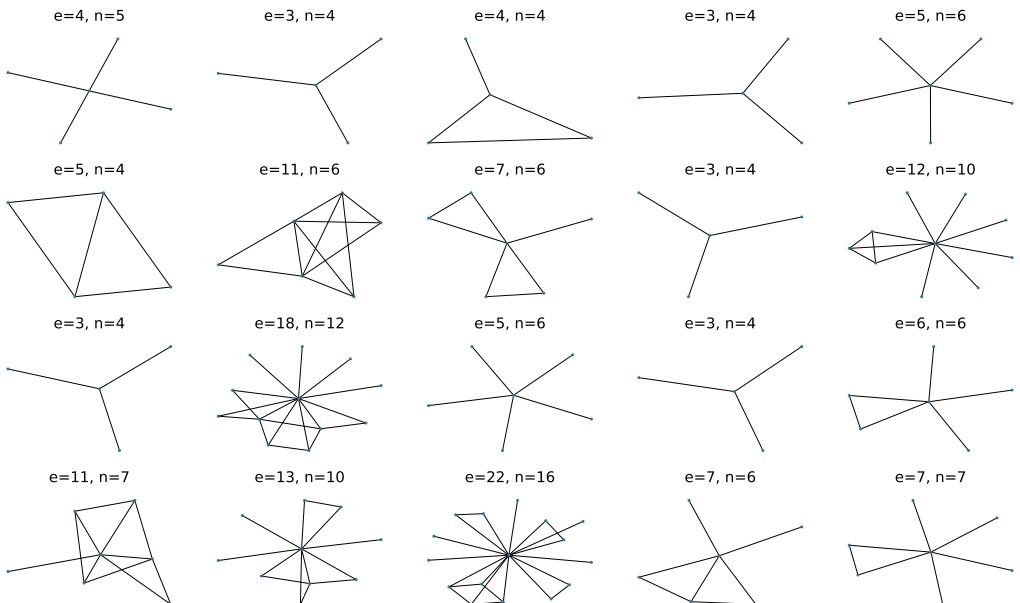

Figure 5: Visualization results on Ego-small dataset.

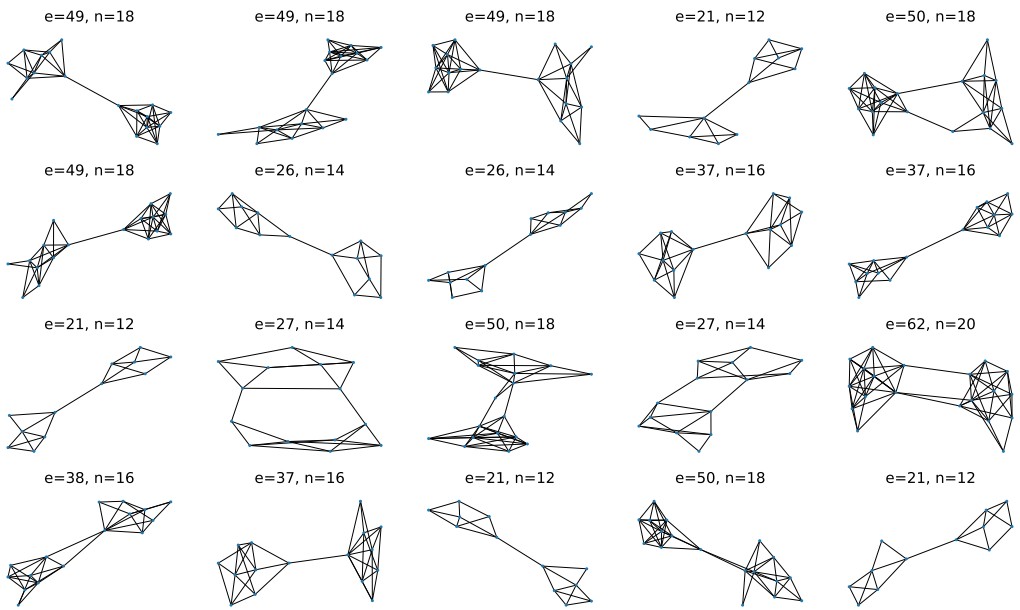

Figure 6: Visualization results on Community-small dataset.

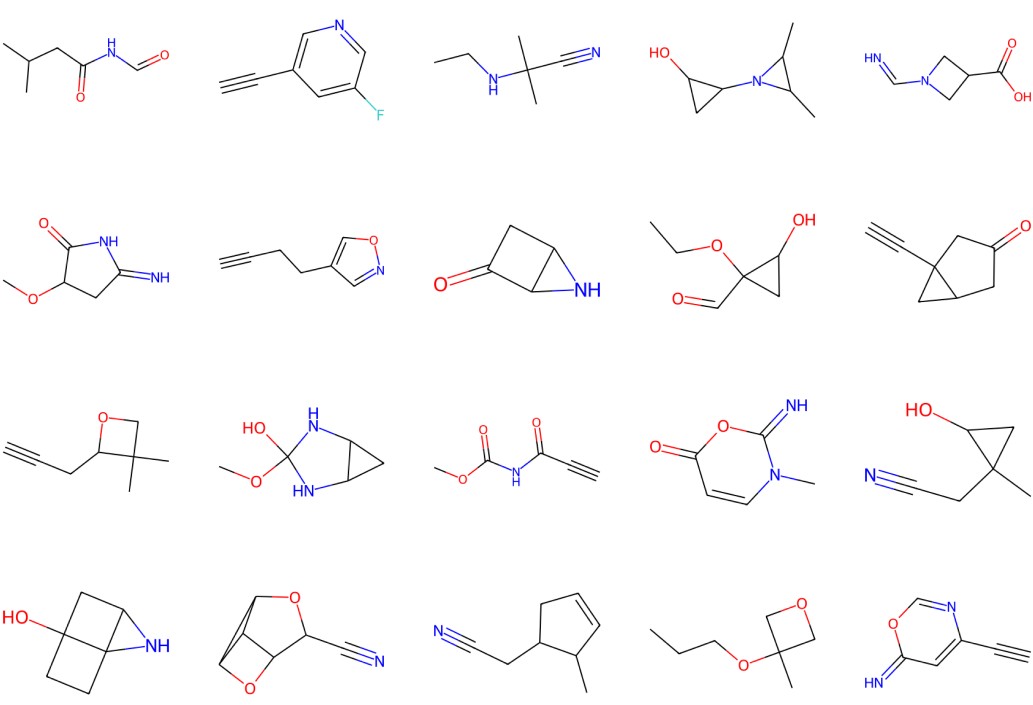

Figure 7: Visualization results on QM9 dataset.

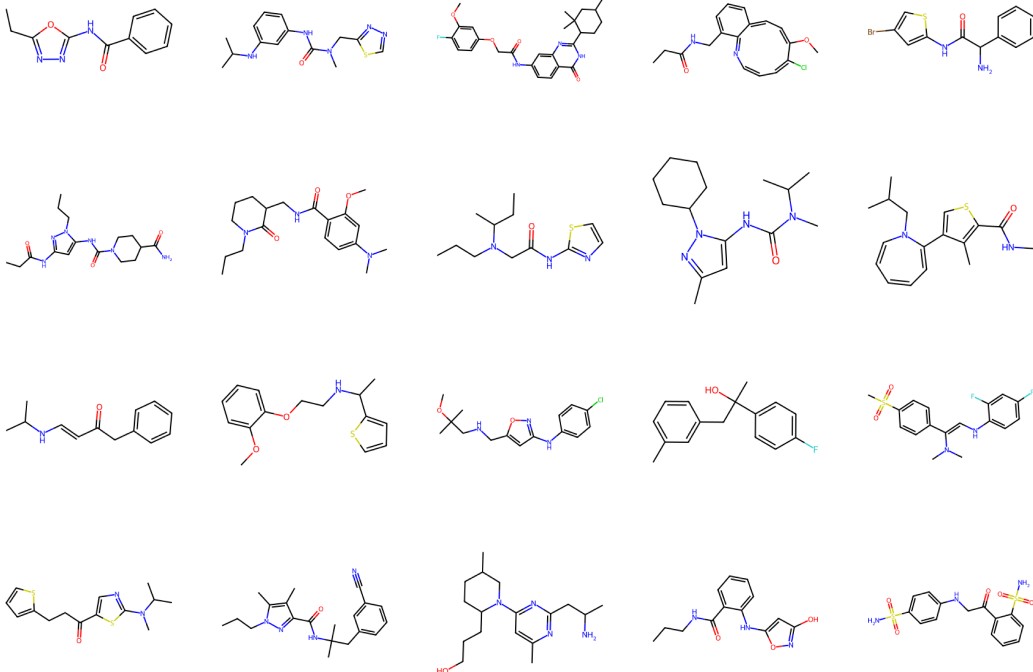

Figure 8: Visualization results on ZINC250k dataset.

