# OpenReview forum: "Global and Local Topology-Aware Graph Generation via Dual Conditioning Diffusion"
_ICLR.cc/2026/Conference — ICLR 2026 Poster_

### Official Review · Reviewer_Cc24 · 2025-10-26

**Soundness:** 2
**Presentation:** 3
**Contribution:** 1
**Rating:** 2
**Confidence:** 4

**Summary:**

This paper proposes DualDiff, a latent diffusion model designed for graph generation that jointly captures both local and global topological information through a two-branch diffusion process and a dual conditioning mechanism.

The model encodes input graphs via a pretrained graph autoencoder, extracts global representations via graph clustering (spectral or K-means), and performs parallel diffusion on node-level and cluster-level latent variables, alternately conditioning one on the other. Experiments are conducted on generic graph datasets (Ego-small, SBM, etc.) and molecular datasets (ZINC250k, QM9, MOSES).

**Strengths:**

- The motivation—capturing multi-scale (global and local) structural information in graph generation—is both timely and important.

- The paper is very well-written and full of details.

- The authors conduct comprehensive experiments, comparing with many recent diffusion-based and autoregressive graph generators.

**Weaknesses:**

1. The paper defines “global information” simply as the cluster centroids obtained from node embeddings via K-means or spectral clustering. This is a very coarse, heuristic, and outdated approximation of global structure. It is not truly “topology-aware” as claimed. The authors do not justify why a fixed, non-learned clustering is preferable.

2. The statement that “existing methods still leverage node-level generative paradigms” is too absolute. Many recent approaches already incorporate hierarchical or subgraph-level generation.

3. Although the authors emphasize that the encoder takes 3D coordinate information as input for molecular graphs, the model ultimately generates only 2D molecular structures (bond graphs) rather than full 3D geometries.

4. The authors claim Eq. (7) “implies that modeling the joint distribution of global and local information can be decomposed as two complementary processes” . In fact, Eq. (7) is merely a standard conditional-probability identity—any joint distribution can be written that way. Therefore, it provides no theoretical evidence that the proposed alternation scheme meaningfully models $p(Z_l​,Z_g​)$.

5. On ZINC250k (Table 2), the reported Validity = 92%, which is much lower than baselines such as GruM (99%), GraphArm (100%), and GDSS (97%).  This indicates a serious failure in generating chemically valid molecules and undermines the claim that DualDiff captures local chemical constraints (e.g., valence, functional groups).

6. Protein datasets, which contain larger graphs (100 < |V| < 500) are also widely used in generic graph generation. Moreover, related baselines such as GruM have reported results on these datasets. The authors are encouraged to include experiments on protein graphs to more comprehensively demonstrate the effectiveness and scalability of their proposed approach.

7. The paper does not provide a code release or supplementary implementation details.
Given the model complexity and multi-stage training (autoencoder + dual diffusion), reproducibility is questionable. Code should be provided for verification.

**Questions:**

1. Why do the authors believe that a fixed, non-learnable clustering method (K-means or spectral) can define “global topology” more effectively than end-to-end hierarchical pooling approaches in modern GNNs?

2. How sensitive are the results to the choice of clustering algorithm and the number of clusters $K$?

3. Why not train the clustering jointly with the autoencoder or in the diffusion process to obtain adaptive global representations?

4. The model’s validity on ZINC250k is only 92%. Does this reflect an inherent inability to capture local chemical rules? Have the authors examined failure cases?

5. Can the framework be extended to explicitly generate 3D molecular structures rather than only 2D topologies?

6. Is there any theoretical basis to justify Eq. (8) as a valid probabilistic factorization of $p(Z_l, Z_g)$?

---

> ### Author Response · Authors · 2025-11-19
>
> Dear Reviewer Cc24,
>
> Thanks for your insightful comments.
>
> **W1**
>
> >Clarification of the clustering methods and comparison between deep clustering.
>
> **Clarification of the clustering methods used**
>
> * We perform clustering in either the 3D geometric space or the topological space of the graph to obtain topologically-based classes. **By combining these geometrically meaningful clusters with node embeddings, we introduce a topological bias**, which facilitates the extraction of effective global information. In general, the geometry or topology of the graph serves as a **powerful, informative, and critical indicator**, further ensuring the effectiveness of the global information extraction.
> * Besides, the simple clustering, such as K-means or spectral clustering, can be **highly effective** for generic and molecular graphs. For example, in Ego graphs, spectral clustering can effectively distinguish the central node from others. In the Community graphs, spectral clustering also achieves almost perfect separation of the two communities. Furthermore, partitioning based on molecular positions can also effectively identify nodes with similar semantics and spatial locations, such as carbon rings and various functional groups.
>
> **Comparison between deep clustering methods**
>
> * Learnable deep graph clustering or pooling methods **extract global information based on the similarity of node embeddings**, typically leveraging K-means clustering or pooling to obtain global representations. The **key difference** between our approach and these methods is that we extract global information without backpropagation or deep training by directly utilizing geometry-based clusters, whereas deep clustering methods depend on the similarity of latent embeddings and **require additional training costs**.
> * We provide a comparison between deep clustering and our approach, as shown below. Notably, K-means and spectral clustering achieve performance comparable to deep clustering, further demonstrating the effectiveness of topological bias in capturing global information. To ensure efficiency and avoid additional training costs, we choose the simple but effective clustering.
>
> | Clustering Methods  | FCD / NSPKD / Unique          |
> | ------------------- | ----------------------------- |
> | K-means             | 0.092 / **0.0001** / 0.95     |
> | Spectral Clustering | 0.094 / 0.0004 / **0.97**     |
> | DiffPool [1]        | 0.096 / 0.0003 / 0.96         |
> | Dink-Net [2]        | **0.090** / **0.0001** / 0.94 |
>
> [1] Ying, Zhitao, et al. Hierarchical graph representation learning with differentiable pooling. NeurIPS, 2018.
>
> [2] Liu, Yue, et al. Dink-net: Neural clustering on large graphs. ICML, 2023.
>
> **W2**
>
> >Statement of the previous graph generative model.
>
> In the original manuscript, our use of "most existing methods" was intended to mean "most of the existing methods", to avoid making an overly absolute statement. Following your insightful suggestion and to further ensure rigor in our expression, we will revise it to: "traditional node-level generative paradigms may have difficulty simultaneously capturing the multiscale dependencies in graphs."
>
> In addition, we also discuss the differences between our model and hierarchical  and subgraph-level generation approaches (in Appendix C.7 and Introduction, respectively). Compared to these advanced subgraph-level and hierarchical methods, our model enables dynamic interaction between global and local information, thereby achieving more effective joint modeling.
>
> **W3**
>
> >Experiments on generating 3D molecules
>
> In the experimental section, our method primarily follows the previous methods and focuses on the 2D graph generation task. The main purpose of incorporating 3D positions is to leverage node distances to implicitly model relationships between nodes, thereby improving the performance of DualDiff.
>
> Our approach can also be naturally extended to generate 3D molecular geometries. Following GEOLDM [3], we replace the lightweight MLP decoder with an EGNN to ensure that both latent embedding and coordinates are simultaneously generated. The experimental results on the QM9 dataset are shown below. Our method achieves competitive performance compared to the advanced GEOLDM and EQUIFM [4], which illustrates the effectiveness of our methods.
>
> | Metrics         | Atom Sta (%) / Mol Sta (%) / Valid & Unique (%) |
> | --------------- | ----------------------------------------------- |
> | Data            | 99.0 / 95.2 / 97.7                              |
> | GEOLDM          | **98.9** / **89.4** / 92.7                      |
> | EQUIFM          | **98.9** / 88.3 / **93.5**                      |
> | DualDiff (Ours) | **98.9** / 88.7 / 93.3                          |
>
> [3] Xu, Minkai, et al. Geometric latent diffusion models for 3d molecule generation. ICLR, 2023.
>
> [4] Song, Yuxuan, et al. Equivariant flow matching with hybrid probability transport for 3d molecule generation. NeurIPS, 2023.

---

> ### Author Response · Authors · 2025-11-19
>
> **W4**
> > Theoretical insights and effectiveness of the alternating mechanism
>
> **Theoretical insights into the alternating mechanism**
>
> To model the high-dimensional and intractable joint probability distribution $P(Z_l, Z_g)$, diffusion-based methods often use the target $Z_{l,0}$ and $Z_{g,0}$ to model the conditional probability path $p_t$. This can be factorized into the following three equivalent forms:
> $$
> \begin{align} p_t(Z_l, Z_g) &= \int_{Z_{l,0}, Z_{g,0} \sim \text{data}} p_t(Z_l, Z_g \mid Z_{l,0}, Z_{g,0}) \mathrm{d}Z_{l,0} \mathrm{d}Z_{g,0} = \int_{Z_{l,0}, Z_{g,0} \sim \text{data}} p_t(Z_l \mid Z_{l,0}) p_t(Z_g \mid Z_l, Z_{g,0}) \mathrm{d}Z_{l,0} \mathrm{d}Z_{g,0} = \int_{Z_{l,0}, Z_{g,0} \sim \text{data}} p_t(Z_l \mid Z_g, Z_{l,0}) p_t(Z_g \mid Z_{g,0}) \mathrm{d}Z_{l,0} \mathrm{d}Z_{g,0} \end{align}
> $$
> These three factorizations provide alternative ways to model the joint conditional probability path $p_t$:
>
> 1. Simultaneously incorporating both global and local conditions.
> 2. Incorporating local information into the generation of global information.
> 3. Incorporating global information into the generation of local information.
>
> Furthermore, in diffusion models, $p_t$ is typically instantiated as a Gaussian path. For example, in the decomposition $p_t(Z_l \mid Z_{l,0}), p_t(Z_g \mid Z_l, Z_{g,0})$, $p_t(Z_l \mid Z_{l,0})$ can be modeled as $\mathcal{N}(Z_l; \mu_t(Z_{l,0}), \sigma_t(Z_{l,0}))$, corresponding to a standard diffusion process **without additional conditions** (i.e., $C_l = 0$ in process (ii)).
>
> Besides, $p_t(Z_g \mid Z_l, Z_{g,0}) = \mathcal{N}(Z_g; \mu_t(Z_l, Z_{g,0}), \sigma_t(Z_l, Z_{g,0}))$, which can be seen as a conditional diffusion process with $Z_l$ as the condition. By further integrating self-conditioning, we obtain $C_ g = (\hat {Z}_ {l,0}, \hat{Z}_ {g,0})$. Similarly, in process (i), $(C_ l, C_ g) = (((\hat{Z}_ {l,0}, \hat{Z}_ {g,0})), 0)$ corresponds to modeling $p_ t(Z_ l \mid Z_g, Z_{l,0})\, p_t(Z_g \mid Z_{g,0})$.
>
>
>
> **Effectiveness of the alternating mechanism**
>
> As previously discussed, the simultaneous and alternating strategies essentially model $p_t$ in different ways. In Appendix D (Table 14), we further **compare the alternating and simultaneous strategies**. The results are also shown below. The learning difficulties of global and local processes are inherently different. When introducing global and local information simultaneously, the quality of the generated local details may be significantly lower than that of the global information. This imbalance may result in **incorrect guidance for the global generation process and thus lead to instability.** In contrast, the alternating strategy enables DualDiff to focus more on the relatively challenging generation of local details, thereby improving both model stability and the convergence of local structures.
>
> | Datasets     | Enzymes                   | Planar                       |
> | ------------ | ------------------------- | ---------------------------- |
> | **Metrics**  | Deg. / Clus. / Orbit      | Deg. / Clus. / Orbit         |
> | alternating  | **0.010 / 0.083 / 0.007** | **0.0003 / 0.0275 / 0.0002** |
> | simultaneous | 0.018 / 0.107 / 0.009     | 0.0009 / 0.0425 / 0.0004     |
>
>
>
> **W5**
>
> >Clarification of the Validity metric.
>
> * Unlike diffusion-based methods that operate directly in the original graph space, our approach is primarily based on latent diffusion methods, which utilize an autoencoder and may introduce **edge reconstruction errors**. This may be an inherent limitation of LGD-based methods. In practice, RDKit tools can be used for **post-processing corrections** on the generated molecules, further ensuring chemical validity.
> * Nevertheless, latent diffusion-based methods DualDiff and EDM-SyCo can fully exploit node features and **demonstrate significant advantages in FCD and KL scores**, which reflect the global distribution.
> * Although DualDiff lags behind advanced models like GruM in terms of the Validity metric, it achieves **substantial improvements in FCD, KL score, and validity** compared to other latent diffusion-based methods such as EDM-SyCo and GEOLDM, as shown below. This further highlights the advantages of the dual conditioning approach in capturing both global and local dependencies within molecular graphs.
>
> | Method   | FCD Score | KL Score | Validity |
> | -------- | --------- | -------- | -------- |
> | GEOLDM   | 0.17      | 0.79     | 0.12     |
> | EDM-SyCo | 0.85      | 0.96     | 0.88     |
> | DualDiff | **0.91**  | **0.98** | **0.92** |

---

> ### Author Response · Authors · 2025-11-19
>
> **W6**
>
> >Experiments on Proteins dataset
>
> Following your insightful suggestion, we have performed more comprehensive evaluations on the Proteins dataset, as shown below. The results provide further evidence of the effectiveness of our proposed approach in large-scale scenarios.
>
> | Methods  | Deg. / Clust. / Orbit / Spec.                  |
> | -------- | ---------------------------------------------- |
> | GDSS     | 0.0861 / 0.5111 / 0.732 / 0.0748               |
> | GruM     | **0.0019** / 0.0660 / 0.0345 / 0.0030          |
> | DualDiff | 0.0022 / **0.0309**  / **0.0067** / **0.0030** |
>
>
>
> **W7**
>
> >Clarification of code and supplementary implementation details.
>
> The architectures and implementation details of all models used in DualDiff are presented in Appendix A, while the experimental settings are provided in Appendix B to facilitate reproducibility.
>
> Furthermore, we plan to release the code if the paper is accepted.
>
>
>
> **Q1**
>
> >Comparison between hierarchical methods and our methods
>
> Unlike end-to-end hierarchical pooling approaches, which extract global information through trainable aggregation of node embeddings, our method employs K-means and spectral clustering to directly leverage geometrically meaningful clusters. This approach requires no backpropagation or deep learning-based training and naturally introduces a geometric bias into the global representation.
>
> As shown in W1, we compare the performance of the advanced DiffPool [1] and simple clustering methods. Notably, K-means and spectral clustering achieve performance comparable to the learnable DiffPool method, while requiring no backpropagation and additional deep training. This further underscores the efficiency and effectiveness of the clustering methods employed in DualDiff.
>
> [1] Ying, Zhitao, et al. Hierarchical graph representation learning with differentiable pooling. NeurIPS, 2018.
>
> **Q2**
>
> >Sensitivity of clustering methods
>
> **Experiments on different clustering methods**
>
> We have conducted comprehensive experiments in Appendix C.5 to assess the impact of different clustering algorithms and the number of clusters on our results, **thereby validating the sensitivity of clustering**. As shown below, the results on the QM9 dataset demonstrate that DualDiff consistently improves over standard latent graph diffusion across various clustering algorithms, which further illustrates the robustness and generalization of our approach.
>
> | Methods             | FCD       | NSPKD      |
> | ------------------- | --------- | ---------- |
> | Standard            | 0.145     | 0.0007     |
> | Spectral Clustering | 0.094     | 0.0004     |
> | GMM                 | 0.095     | 0.0003     |
> | K-means             | **0.092** | **0.0001** |
>
> **Experiments on different numbers of clusters**
>
> We further evaluate the model performance under different numbers of clusters. The results are shown below. We report the average of Deg., Clus. and Orbit metrics. All tested cluster numbers lead to performance improvements compared to standard latent graph diffusion, which further demonstrates the robustness and generalization of our algorithm.
>
> | Methods           | Enzymes    | Planar     |
> | ----------------- | ---------- | ---------- |
> | Standard          | 0.0672     | 0.0143     |
> | DualDiff ($K=4$)  | 0.0443     | 0.0104     |
> | DualDiff ($K=8$)  | **0.0333** | 0.0064     |
> | DualDiff ($K=10$) | 0.0353     | **0.0056** |

---

> > ### Author Response · Authors · 2025-11-19
> >
> > **Q3**
> >
> > >Clarification of jointly training AutoEncoder or diffusion with clustering.
> >
> > **Train clustering jointly with Autoencoder**
> >
> > * Jointly training the clustering with the autoencoder is essentially **a form of deep graph clustering**, where global information is derived from the similarity of latent node embeddings. However, in generic graphs with limited node features (typically limited to basic attributes such as node degree), this approach **may not effectively capture topological information**. For example, in community graphs, it may struggle to distinguish nodes from different communities if they have the same degree.
> > * In contrast, our method for global information extraction directly introduces bias from the underlying geometry or topology of the graph, thereby more effectively capturing structural characteristics.
> > * We further compare the performance between these two approaches, as shown below. We report the average of Deg., Clust. and Orbit metrics. The results largely underscore the importance of introducing geometric bias in global information extraction.
> >
> > | Methods             | Ego-Small | Community-small | Grid       |
> > | ------------------- | --------- | --------------- | ---------- |
> > | Seperate (original) | **0.005** | **0.007**       | **0.0003** |
> > | Joint with AutoEnc. | 0.010     | 0.012           | 0.0006     |
> >
> > **Train clustering jointly with Diffusion**
> >
> > * Training the diffusion model and clustering methods jointly may lead to instability and convergence issues, ultimately affecting model performance. In latent diffusion models, the autoencoder and diffusion modules are typically trained in a two-stage manner, which is widely recognized as beneficial for improving convergence and final performance. This is also why we first extract local and global representations and then train the diffusion model separately.
> > * We further conducted experiments comparing joint training and separate (two-stage) training, as shown below. The results demonstrate the effectiveness of the two-stage training strategy. Additionally, in dual conditioning, the global information of local details can be dynamically extracted according to the association matrix between global and local representations, which ensures both dynamic and efficiency.
> >
> > | Methods             | Ego-Small | Community-small | Grid       |
> > | ------------------- | --------- | --------------- | ---------- |
> > | Seperate (original) | **0.005** | **0.007**       | **0.0003** |
> > | Joint with Diff.    | 0.014     | 0.021           | 0.0011     |
> >
> > **Q4**
> >
> > >Clarification of Validity and failure cases.
> >
> > **Validity.** As described in W5, compared to diffusion paradigms that directly operate in the original graph space and jointly model edges and nodes, the autoencoder used in latent graph diffusion (LGD) approaches may introduce additional edge reconstruction errors, which may be an inherent limitation of LGD-based methods. However, compared with other LGD-based approaches such as EDM-SyCo and GEOLDM, DualDiff achieves further improvements in validity, FCD, and KL metrics, demonstrating the advantage of dual conditioning in simultaneously capturing both local and global information.
> >
> > **Failure cases.** The failure cases are mainly due to Kekulization errors and valency errors, which result from inconsistent bond type predictions. In practical applications, RDKit tools can be utilized for **post-processing corrections** on the generated molecules, further ensuring the chemical validity.
> >
> >
> >
> > **Q5**
> >
> > >Extension to 3D molecular generation
> >
> > As described in W3, our method can be naturally extended to 3D molecular generation and demonstrates competitive performance compared to state-of-the-art approaches such as GEOLDM and EQUIFM.
> >
> >
> >
> > **Q6**
> >
> > >Clarification of Equ.8
> >
> > As described in W4, in process (i), the conditions are set as $(C_ l, C_ g) = ((\hat{Z}_ {l,0}, \hat{Z}_ {g, 0}), 0)$. Here, $C_ g = 0$ corresponds to the standard diffusion process without additional conditioning to model $p_ t(Z_ g \mid Z_ {g, 0}) = \mathcal{N}(Z_ g; \mu_ t(Z_ {g, 0}), \sigma_ t(Z_ {g, 0}))$. On the other hand, $C_ l$ introduces both self-conditioning and $\hat{Z}_ {g,0}$, and thus models $p_ t(Z_ l \mid Z_ g, Z_ {l,0}) = \mathcal{N}(Z_ l; \mu_ t(Z_ g, Z_ {l,0}), \sigma_ t(Z_ g, Z_ {l,0})).$ Here, we use $Z_ g = \hat{Z}_ {g,0}$ rather than $Z_ {g,t}$. Because $Z_ {g,t}$ is typically noisy, using the estimated $\hat{Z}_ {g,0}$ can provide more reliable guidance during generation. Therefore, in the alternating mechanism, process (i)  essentially models $p_ t(Z_ g \mid Z_ {g,0})$ and $p_ t(Z_ l \mid Z_ g, Z_ {l,0})$. Similarly, in process (ii), the conditioning is set up to model $p_ t(Z_ l \mid Z_ {l,0})$ and $p_t(Z_ g \mid Z_ l, Z_ {g,0})$, respectively.
> >
> >
> >
> > We hope our replies address all your concerns. If you have any further questions, please do not hesitate to let us know.

---

> > > ### Author Response · Authors · 2025-11-26
> > > **Looking forward to your reply**
> > >
> > > Dear Reviewer Cc24,
> > >
> > > We sincerely appreciate the time and effort you have dedicated to reviewing our paper. Given the limited timeframe for author–reviewer discussion, **we kindly ask whether our responses have addressed your concerns**.
> > >
> > > Following your valuable suggestions, we have improved the paper in the following aspects:
> > >
> > > * We have compared the clustering methods used for global information extraction with advanced learnable clustering approaches to demonstrate their efficiency and effectiveness. In addition, we further discussed the sensitivity of the clustering methods to demonstrate the robustness of our approach.
> > > * Additionally, we have included experiments on 3D molecular generation and protein datasets to further demonstrate the effectiveness of our method.
> > > * We have provided additional theoretical insights into the alternating mechanism and further analyzed its advantages compared to the simultaneous strategy.
> > > * We have conducted a more detailed analysis of the Validity metric on ZINC250K and demonstrated the advantages of DualDiff over other latent diffusion-based methods.
> > >
> > > Thanks again for your insightful review. We look forward to your feedback.

---

### Official Review · Reviewer_rfZi · 2025-10-31

**Soundness:** 3
**Presentation:** 3
**Contribution:** 3
**Rating:** 6
**Confidence:** 3

**Summary:**

This paper proposes a latent graph diffusion model that concurrently denoises node-level and cluster-level representations, coupled with a bidirectional conditioning mechanism to exchange global and local topological cues during generation. Extensive experiments on eight generic and molecular graph benchmarks demonstrate competitive or state-of-the-art performance in both unconditional and property-conditional generation while requiring fewer diffusion steps than prior diffusion counterparts.

**Strengths:**

1. The paper is well-written and easy to follow.
2. Using global & local latent features in graph diffusion models with learnable cross-conditioning is a reasonable design for the task.
3. The empirical results across diverse datasets show the effectiveness of the proposed method.

**Weaknesses:**

1. The paper uses different global feature extraction methods in different tasks, but how to choose the method is unclear, which may lead to difficulty in generalization to new tasks.
2. In ablation study, what if there is only local-to-global condition?
3. At the beginning of sampling process, the number of nodes and clusters should be pre-determined, how to decide these numbers? What about the generalization ability of these numbers?

**Questions:**

N/A

---

> ### Author Response · Authors · 2025-11-19
>
> Dear Reviewer rfZi,
>
> Thanks for your insightful comments.
>
>
>
> **W1**
>
> >The generalization of different clustering methods and how to choose clustering methods
>
> **Clustering methods**
>
> We have conducted comprehensive experiments in Appendix C.5 to assess the impact of different clustering algorithms on our results, thereby validating the generalization of our methods.  As shown below, the results on the QM9 dataset demonstrate that DualDiff consistently achieves improvements over the standard latent graph diffusion across various clustering algorithms, which fully illustrates the robustness and generalization of our approach.
>
> | Methods             | FCD       | NSPKD      |
> | ------------------- | --------- | ---------- |
> | Standard            | 0.145     | 0.0007     |
> | Spectral Clustering | 0.094     | 0.0004     |
> | GMM                 | 0.095     | 0.0003     |
> | K-means             | **0.092** | **0.0001** |
>
>
>
> **The choice of clustering methods**
>
> Regarding the choice of clustering methods, for generic graphs containing only topological information, spectral clustering is highly effective at detecting different topological structures. For example, in Ego graphs, spectral clustering can effectively distinguish the central node from the others. In the Community-small dataset, spectral clustering also achieves almost perfect separation of the two communities.
>
> For molecular graphs, besides clustering methods such as K-means, spectral clustering, GMM, and the Louvain algorithm, we also experiment with advanced deep clustering methods, such as DiffPool [1] and Dink-Net [2]. The results on the QM9 dataset are shown below. Compared to deep clustering methods, the K-means algorithm achieves comparable performance without additional training overhead, demonstrating both effectiveness and efficiency. Therefore, we choose K-means as the clustering method for molecular graphs.
>
> | Clustering Methods | FCD / NSPKD / Unique                    |
> | ------------------ | --------------------------------------- |
> | K-means (original) | <u>0.092</u> / **0.0001** / <u>0.95</u> |
> | DiffPool           | 0.096 / 0.0003 / **0.96**               |
> | Dink-Net           | **0.090** / **0.0001** / 0.94           |
>
> [1] Ying, Zhitao, et al. Hierarchical graph representation learning with differentiable pooling. NeurIPS, 2018.
>
> [2] Liu, Yue, et al. Dink-net: Neural clustering on large graphs. ICML, 2023.
>
>
>
> **W2**
>
> >Ablation study of introducing local-to-global only.
>
> Following your insightful suggestion, we added an ablation study on the local-to-global condition, as shown below. This further demonstrates the necessity of the dual conditioning mechanism for achieving dynamic interaction between global and local information. We will incorporate this part into the updated version of our paper.
>
> | Methods                             | FCD score       | KL score        |
> | ----------------------------------- | --------------- | --------------- |
> | DualDiff (w/o any Cond.)            | 0.65 $\pm$ 0.01 | 0.82 $\pm$ 0.02 |
> | DualDiff ($Z_l\rightarrow Z_g$)     | 0.75 $\pm$ 0.03 | 0.95 $\pm$ 0.02 |
> | DualDiff ($Z_l\leftrightarrow Z_g$) | 0.91 $\pm$ 0.02 | 0.98 $\pm$ 0.01 |
>
>
>
> **W3**
>
> >How to determine the number of nodes and clusters.
>
> **Number of nodes**
>
> Regarding the number of nodes, we follow previous work by randomly sampling from the distribution of node numbers in the training set. This is a widely adopted approach in the field of graph generation.
>
> **Number of clusters**
>
> For the number of clusters, we typically select from the range {6, 8, 10} in our experiments, which ensures both efficiency and accurate extraction of global information. To further validate the generalizability of the cluster number, we evaluate the model performance under different numbers of clusters. The results are shown below. We report the average of Deg., Clus. and Orbit metrics. Compared to the baseline without global information, all tested cluster numbers lead to performance improvements, which further demonstrates the robustness and generalization of our algorithm.
>
> | Methods           | Enzymes    | Planar     |
> | ----------------- | ---------- | ---------- |
> | Baseline          | 0.0672     | 0.0143     |
> | DualDiff ($K=4$)  | 0.0443     | 0.0104     |
> | DualDiff ($K=8$)  | **0.0333** | 0.0064     |
> | DualDiff ($K=10$) | 0.0353     | **0.0056** |
>
> We hope our replies address all your concerns. If you have any further questions, please do not hesitate to let us know.

---

> > ### Author Response · Authors · 2025-11-26
> > **Looking forward to your reply**
> >
> > Dear Reviewer rfZi,
> >
> > We sincerely appreciate the time and effort you have dedicated to reviewing our paper. Given the limited timeframe for author–reviewer discussion, **we kindly ask whether our responses have addressed your concerns**.
> >
> > Following your valuable suggestions, we have improved the paper in the following aspects:
> >
> > - We have analyzed the sensitivity of the clustering method to further demonstrate the robustness and effectiveness of our approach.
> > - We have added the ablation study of the local-to-global condition, which further illustrates the effectiveness of the dual conditioning mechanism.
> > - We have analyzed how to determine the number of nodes and clusters, further ensuring the generalizability of our method.
> >
> > Thanks again for your insightful review. We look forward to your feedback.

---

### Official Review · Reviewer_o5es · 2025-11-01

**Soundness:** 3
**Presentation:** 3
**Contribution:** 2
**Rating:** 6
**Confidence:** 2

**Summary:**

This paper proposes DualDiff, a dual-branch latent diffusion model that learns both global and local graph structures. The key idea is to run two diffusion processes, one at the node level and one at the cluster level, and let them interact through a dual conditioning mechanism. This design helps the model capture both fine-grained details and overall topology. Experiments on multiple benchmarks show clear gains over prior graph diffusion models, proving that the method effectively models multi-scale dependencies in graphs.

**Strengths:**

1. Good motivation and clear design idea
The paper starts from a real problem that most graph diffusion models only handle node-level stuff. Splitting the process into local and global branches makes sense and is well explained.
2. Thoughtful architectural design
Alternating between the two diffusion branches is clever. It’s like doing local updates and then global aggregation, which keeps the training stable and avoids conflicts between the two processes.
3. Strong and broad experiments
They test on both synthetic graphs and real molecular datasets, and the gains are consistent. It shows the method isn’t overfitted to one type of data.

**Weaknesses:**

1. too dependent on clustering
The global branch comes from clustering nodes, so if the clustering is poor, the “global info” might be misleading. They don’t analyze this sensitivity much.
2 . Efficiency claim not fully convincing
They say the model is efficient in the latent space, but with two diffusion networks and alternating steps, it’s unclear how much heavier it actually is.
3. No solid theoretical grounding for stability
The alternating process is explained intuitively (like server-client updates), but there’s no formal guarantee or analysis of convergence.

**Questions:**

1. The paper mentions that the alternating scheme improves stability, but is there any quantitative or theoretical evidence to support this? What happens if the two processes are trained simultaneously, rather than alternately?
2. How sensitive is the model to the choice of clustering algorithm or the number of clusters K? Did you try other global extraction methods besides K-means or spectral clustering?
3. Computation and scalability: Since you have two denoising networks and alternating updates, how much additional compute or memory does this introduce compared to a standard latent diffusion model, such as Latent Graph Diffusion?

---

> ### Author Response · Authors · 2025-11-19
>
> Dear Reviewer o5es,
>
> Thanks for your insightful comments.
>
> **W1**
>
> >Sensitivity analysis of clustering methods.
>
> **Experiments on different clustering methods**
>
> We have conducted comprehensive experiments in Appendix C.5 to assess the impact of different clustering algorithms and the number of clusters on our results, **thereby validating the sensitivity of clustering**. As shown below, the results on the QM9 dataset demonstrate that DualDiff consistently achieves improvements over the standard latent graph diffusion across various clustering algorithms, which fully illustrates the robustness and generalization of our approach.
>
> | Methods             | FCD       | NSPKD      |
> | ------------------- | --------- | ---------- |
> | Standard            | 0.145     | 0.0007     |
> | Spectral Clustering | 0.094     | 0.0004     |
> | GMM                 | 0.095     | 0.0003     |
> | K-means             | **0.092** | **0.0001** |
>
> **Experiments on different numbers of clusters**
>
> We further evaluate the model performance under different numbers of clusters. The results are shown below. We report the average of Deg., Clus. and Orbit metrics. All tested cluster numbers lead to performance improvements compared to standard latent graph diffusion, which further demonstrates the robustness and generalization of our algorithm.
>
> | Methods           | Enzymes    | Planar     |
> | ----------------- | ---------- | ---------- |
> | Standard          | 0.0672     | 0.0143     |
> | DualDiff ($K=4$)  | 0.0443     | 0.0104     |
> | DualDiff ($K=8$)  | **0.0333** | 0.0064     |
> | DualDiff ($K=10$) | 0.0353     | **0.0056** |
>
> Additionally, the clustering algorithms of global information extraction are all directly based on the graph structure or geometric space. In general, the geometry of a graph serves as a powerful, informative, and important indicator, which further ensures the effectiveness of global information extraction.
>
>
>
> **W2**
>
> >The overhead of dual conditioning and the additional diffusion process.
>
> **Complexity Analysis**
>
> * First, the proposed DualDiff operates at the node and cluster levels, with a diffusion space complexity of $O(N)$. This is significantly more efficient than traditional graph generation paradigms, which require $O(N^2)$ complexity to explicitly model edge information.
> * Although our method introduces additional dual conditioning and an extra diffusion process to capture global information, **the number of clusters is generally much smaller than the number of nodes**, i.e., $K$ << $N$. Therefore, the additional global diffusion process does not incur much overhead.
> * Moreover, the alternating mechanism within the dual conditioning can essentially be regarded as an extension of the self-conditioning method for modeling joint probability distributions. As clearly stated in the Self-Conditioning paper [1], this mechanism "comes at a negligible extra time and memory cost during sampling, and the overall increase in training time is small (e.g., less than 25%)." Besides, there is no expensive backpropagation during sampling, further guaranteeing the efficiency of alternating steps.
>
> **Practical Overhead**
>
> To further validate the efficiency of our model, we measured the **practical sampling time and memory**. The results on QM9 datasets are shown below, where "Baseline" denotes standard latent graph diffusion. As can be seen from the table, the additional time and memory cost introduced by our model is acceptable, which further guarantees the efficiency of our methods.
>
> | Methods            | Time (s) | Memory (MB) |
> | ------------------ | -------- | ----------- |
> | Baseline           | 29       | 2465        |
> | DualDiff w/o dual. | 32       | 2826        |
> | DualDiff           | 41       | 3026        |
>
> [1] Chen, T. et al. Analog bits: Generating discrete data using diffusion models with self-conditioning. ICLR, 2023.

---

> ### Author Response · Authors · 2025-11-19
>
> **W3**
>
> >Analysis of the stability and convergence of the alternating mechanism.
>
> **Stability**
>
> In Appendix D, we compare the alternating strategy with the symmetric strategy, which incorporates global and local information **simultaneously**, as shown below. Essentially, the learning difficulty of global and local processes is distinct. When introducing global and local conditions simultaneously, the quality of the generated local details may be **significantly lower than** that of the global information. This imbalance may result in **incorrect guidance** for the global generation and lead to **instability**. In contrast, the alternating strategy encourages DualDiff to **focus more on the challenging generation of local details**, thereby enhancing both the model’s stability and the convergence of local structures.
>
> | Datasets     | Enzymes                   | Planar                       |
> | ------------ | ------------------------- | ---------------------------- |
> | **Metrics**  | Deg. / Clus. / Orbit      | Deg. / Clus. / Orbit         |
> | alternating  | **0.010 / 0.083 / 0.007** | **0.0003 / 0.0275 / 0.0002** |
> | simultaneous | 0.018 / 0.107 / 0.009     | 0.0009 / 0.0425 / 0.0004     |
>
> **Convergence**
>
> With respect to convergence, the global information **provides a "big picture" for the generation of local details, facilitating convergence and constraining the exploration space for local details**. We further compare the performance of simultaneous and alternating methods under the same number of sampling steps. The results show that our approach can achieve superior performance with a much smaller number of steps.
>
> | Datasets        | Methods      | 50        | 100       | 200       |
> | --------------- | ------------ | --------- | --------- | --------- |
> | Community-small | Simultaneous | 0.107     | 0.068     | 0.016     |
> |                 | Alternating  | **0.098** | **0.043** | **0.009** |
> | Ego-small       | Simultaneous | 0.044     | 0.016     | 0.007     |
> |                 | Alternating  | **0.032** | **0.012** | **0.004** |
>
>
>
> **Q1**
>
> >Analysis of the stability of the alternating mechanism.
>
> As described in W2, we compare the alternating and simultaneous strategies, which demonstrate the effectiveness of the alternating approach. Essentially, the number $K$ of global information components is much smaller than the number $N$ of local details, making their learning relatively easier. The alternating strategy encourages DualDiff to focus more on the relatively challenging generation of local details, thereby promoting both the stability of the model and the convergence of local details.
>
>
>
> **Q2**
>
> >Sensitivity of clustering methods and  other global extraction methods.
>
> **Clustering methods and the number of clusters $K$**
>
> In W1 and Appendix C.5, we investigate the impact of different clustering methods and cluster numbers on model performance. The results demonstrate that our method remains effective across various clustering methods, further highlighting its robustness and effectiveness. In addition, experiments with different cluster numbers also illustrate the generalization of our methods.
>
> **Other global extraction methods**
>
> In practice, besides simple clustering such as K-means, spectral clustering, GMM, and the Louvain algorithm, we also experimented with trainable deep clustering methods. The results on the QM9 dataset are shown below. Compared to deep clustering methods, the K-means algorithm achieves comparable performance without additional training overhead, further demonstrating the effectiveness and efficiency of our clustering approach.
>
> | Clustering Methods | FCD / NSPKD / Unique                    |
> | ------------------ | --------------------------------------- |
> | K-means (original) | 0.092 / **0.0001** / 0.95 |
> | DiffPool [2]       | 0.096 / 0.0003 / **0.96**               |
> | Dink-Net [3]       | **0.090** / **0.0001** / 0.94           |
>
> [2] Ying, Zhitao, et al. Hierarchical graph representation learning with differentiable pooling. NeurIPS, 2018.
>
> [3] Liu, Yue, et al. Dink-net: Neural clustering on large graphs. ICML, 2023.
>
> **Q3**
>
> >Additional compute or memory for DualDiff.
>
> As described in W3, we evaluated the practical sampling efficiency (in terms of time and memory) of DualDiff compared to standard latent diffusion. Since the number of clusters is much smaller than the number of local nodes $N$, the computational overhead of modeling the diffusion process for global information is significantly lower than that for local details. Furthermore, in the dual conditioning mechanism, the calculation of the affiliation matrix between global and local information has a complexity of $O(N \times K)$. Given that $K \ll N$, this can be approximately regarded as linear complexity and will not introduce much overhead.
>
>
>
> We hope our replies address all your concerns. If you have any further questions, please do not hesitate to let us know.

---

> > ### Author Response · Authors · 2025-11-26
> > **Looking forward to your reply**
> >
> > Dear Reviewer o5es,
> >
> > We sincerely appreciate the time and effort you have dedicated to reviewing our paper. Given the limited timeframe for author–reviewer discussion, **we kindly ask whether our responses have addressed your concerns**.
> >
> > Following your valuable suggestions, we have improved the paper in the following aspects:
> >
> > - We have analyzed the sensitivity of the clustering method to further demonstrate the robustness and effectiveness of our approach.
> > - We have discussed the cost of the dual conditioning mechanism in terms of complexity and practical overhead to illustrate the efficiency of our method.
> > - We have further analyzed the stability and convergence of the alternating method to ensure its soundness.
> >
> > Thanks again for your insightful review. We look forward to your feedback.

---

### Meta-Review · Area_Chair_yVJc · 2025-12-30

**Summary:**

## **Summary of the Paper**

This paper proposes DualDiff, a latent diffusion framework for graph generation that explicitly models both local (node-level) and global (cluster-level) topology. The method introduces a two-branch diffusion process operating on node and cluster representations, coupled via a dual conditioning mechanism that alternates between global-to-local and local-to-global guidance. The goal is to better capture multiscale dependencies in sparse graphs. Extensive experiments on generic graph benchmarks and molecular datasets (e.g., ZINC250k, QM9, MOSES) demonstrate improved performance over prior graph diffusion and autoregressive baselines, particularly in global distribution metrics and efficiency.

## **Reviewer Concerns Before Rebuttal**

Reviewers raised several substantive concerns:

- Dependence on clustering quality: The global branch relies on non-learned clustering (K-means / spectral), raising questions about robustness, sensitivity to the number of clusters, and whether this constitutes a sufficiently “topology-aware” notion of global structure.

- Theoretical justification: The probabilistic interpretation of the alternating dual conditioning  was viewed as largely heuristic, with limited formal guarantees for stability or convergence.

- Computational overhead and scalability: Concerns were raised about the cost of maintaining two diffusion networks and alternating updates, and whether the efficiency claims were convincing.

- Empirical gaps: Specific issues included relatively lower validity on ZINC250k compared to strong autoregressive baselines, lack of protein-scale experiments, unclear ablations (e.g., one-directional conditioning), and questions about reproducibility and code availability.


## **How the Rebuttal Addressed the Concerns**

The rebuttal made a substantial effort to address most concerns:

- Clustering sensitivity and generalization: New ablations across clustering methods (K-means, spectral, GMM, Louvain, and deep clustering) and varying cluster numbers show consistent gains, suggesting robustness to clustering choices.

- Alternating mechanism: Additional empirical comparisons between alternating and simultaneous conditioning demonstrate improved stability and faster convergence. The authors also clarified the probabilistic interpretation and positioned the method as an efficient conditional factorization rather than a strict theoretical guarantee.

- Efficiency and scalability: Detailed complexity analysis and empirical timing/memory results indicate that the additional overhead is modest due to the small number of clusters relative to nodes.

- Expanded experiments: New results on protein datasets and 3D molecular generation were added, strengthening the empirical scope. Validity concerns were contextualized relative to other latent diffusion baselines, with analysis of failure cases.

- Clarity and ablations: Additional ablation studies (e.g., local-to-global only) and implementation details were provided, improving transparency and reproducibility (with a code release promised upon acceptance).

Overall, while parts of the theoretical justification remain heuristic, the rebuttal convincingly addresses the main empirical and methodological concerns, substantially strengthening the submission. Moreover, the majority of reviewers view the paper positively; in my judgment, Reviewer Cc24’s assessment is somewhat harsher than warranted given the added evidence and clarifications. I therefore recommend acceptance.

**Reviewer Concerns:**

See summary above.

**Reviewer Scores:**

I would expect Reviewers rfZi and o5es to remain positive or potentially increase their scores. Given the comprehensive rebuttal and additional experiments, I would also expect Reviewer Cc24 to update their assessment in a more favorable direction.

---

### Decision · Program_Chairs · 2026-01-26

Accept (Poster)